# Nonlinear sensitivity of glacier mass balance to future climate change unveiled by deep learning

Jordi Bolibar [1,2,3 ✉], Antoine Rabatel [1], Isabelle Gouttevin[4], Harry Zekollari[5,6] & Clovis Galiez[7]

Glaciers and ice caps are experiencing strong mass losses worldwide, challenging water availability, hydropower generation, and ecosystems. Here, we perform the first-ever glacier evolution projections based on deep learning by modelling the 21st century glacier evolution in the French Alps. By the end of the century, we predict a glacier volume loss between 75 and 88%. Deep learning captures a nonlinear response of glaciers to air temperature and precipitation, improving the representation of extreme mass balance rates compared to linear statistical and temperature-index models. Our results confirm an over-sensitivity of temperature-index models, often used by large-scale studies, to future warming. We argue that such models can be suitable for steep mountain glaciers. However, glacier projections under low-emission scenarios and the behaviour of flatter glaciers and ice caps are likely to be biased by mass balance models with linear sensitivities, introducing long-term biases in sea-level rise and water resources projections.

[1] Univ. Grenoble Alpes, CNRS, IRD, G-INP, Institut des Géosciences de l'Environnement, Grenoble, France. [2] INRAE, UR RiverLy, Lyon-Villeurbanne, France. [3] Institute for Marine and Atmospheric research Utrecht, Utrecht University, Utrecht, Netherlands. [4] Univ. Grenoble Alpes, Université de Toulouse, Météo-France, CNRS, CNRM, Centre d'Études de la Neige, Grenoble, France. [5] Department of Geoscience and Remote Sensing, Delft University of Technology, Delft, Netherlands. [6] Laboratoire de Glaciologie, Université Libre de Bruxelles, Brussels, Belgium. [7] Univ. Grenoble Alpes, CNRS, G-INP, Laboratoire Jean Kuntzmann, Grenoble, France. ✉email: j.bolibar@uu.nl

Glaciers are experiencing important changes throughout the world as a consequence of anthropogenic climate change[1]. Despite marked differences among regions, the generalized retreat of glaciers is expected to have major environmental and social impacts[2,3]. Water resources provided by glaciers sustain around 10% of the world's population living near mountains and the contiguous plains[4], depending on them for agriculture, hydropower generation[5], industry or domestic use. Several aquatic and terrestrial ecosystems depend on these water resources as well, which ensure a base runoff during the warmest or driest months of the year[6]. Predicting future glacier evolution is of paramount importance in order to correctly anticipate and mitigate the resulting environmental and social impacts. During the last decade, various global glacier evolution models have been used to provide estimates on the future sea-level contribution from glaciers[7,8]. All these glacier models, independently from their approach, need to resolve the two main processes that determine glacier evolution: (1) glacier mass balance, as the difference between the mass gained via accumulation (e.g. snowfall, avalanches and refreezing) and the mass lost via different processes of ablation (e.g. melt and sublimation of ice, firn and snow; or calving)[9]; and (2) ice flow dynamics, characterized by the downward movement of ice due to the effects of gravity in the form of deformation of ice and basal sliding. Simulating these processes at a large geographical scale is challenging, with models requiring several parametrizations and simplifications to operate. Recent efforts have been made to improve the representation of ice flow dynamics in these models, replacing empirical parametrizations with simplified physical models[9,10]. Nonetheless, to represent the glacier mass balance, the vast majority of large-scale glacier evolution models relies on temperature-index models. This type of model uses a calibrated linear relationship between positive degree-days (PDDs) and the melt of ice or snow[11]. The main reason for their success comes from their suitability to large-scale studies with a low density of observations, in some cases displaying an even better performance than more complex models[12]. However, both the climate and glacier systems are known to react non-linearly, even to pre-processed forcings like PDDs[13], implying that these models can only offer a linearized approximation of climate-glacier relationships. Deep artificial neural networks (ANNs) are nonlinear models that offer an alternative approach to these classic methods. However, the use of ANNs remains largely unexplored in glaciology for regression problems, with only a few studies using shallow ANNs for predicting the ice thickness[14] or mass balance[13] of a single glacier.

The French Alps, located in the westernmost part of the European Alps, experience some of the strongest glacier retreat in the world[15–17]. Long-term historical interactions between French society and glaciers have developed a dependency of society on them for water resources, agriculture, tourism[18]—particularly the ski business[19]—and hydropower generation. This rapid glacier retreat is already having an environmental impact on natural hazards[20], mountain ecosystems[21], and biodiversity[6]. Without these cold water resources during the hottest months of the year, many aquatic and terrestrial ecosystems will be impacted due to changes in runoff, water temperature or habitat humidity[6,21,22]. An accurate prediction of future glacier evolution will be crucial to successfully adapt socioeconomic models and preserve biodiversity. Glaciers in the European Alps have been monitored for several decades, resulting in the longest observational series in the world[23,24]. In this study, we investigate the future evolution of glaciers in the French Alps and their nonlinear response to multiple climate scenarios. We perform, to the best of our knowledge, the first-ever deep learning (i.e. deep artificial neural networks) glacier evolution projections by modelling the regional evolution of French alpine glaciers through the 21st century. For this, a newly-developed state-of-the-art modelling framework based on a deep learning mass balance component and glacier-specific parametrizations of glacier changes is used. The climatic forcing comes from high-resolution climate ensemble projections from 29 combinations of global climate models (GCMs) and regional climate models (RCMs) adjusted for mountain regions for three Representative Concentration Pathway (RCP) scenarios: 2.6, 4.5, and 8.5[25]. The 29 RCP-GCM-RCM combinations available, hereafter named climate members, are representative of future climate trajectories with different concentration levels of greenhouse gases (Table S1). With this study, we provide new predictions of glacier evolution in a highly populated mountain region, while investigating the role of nonlinearities in the response of glaciers to multiple future climate forcings.

## Results

**Glacier evolution through the 21st century.** Our projections show a strong glacier mass loss for all 29 climate members, with average ice volume losses by the end of the century of 75%, 80%, and 88% compared to 2015 under RCP 2.6 ($\pm$9%, $n = 3$), RCP 4.5 ($-17\%$ $+11\%$, $n = 13$) and RCP 8.5 ($-15\%$ $+11\%$, $n = 13$), respectively (Fig. 1 and S1). Differences in projected glacier changes become more pronounced from the second half of the century, when about half of the initial 2015 ice volume has already been lost independent of the considered scenario. Annual glacier-wide mass balance (MB) is estimated to remain stable at around $-1.2$ m.w.e. a$^{-1}$ throughout the whole century under RCP 4.5, with glacier retreat to higher elevations (positive effect on MB) compensating for the warmer climate (negative effect on MB). Conversely, for RCP 8.5, annual glacier-wide MB are estimated to become increasingly negative by the second half of the century, with average MB almost twice as negative as today's average values (Fig. 1a). MB rates only begin to approach equilibrium towards the end of the century under RCP 2.6, for which glaciers could potentially stabilize with the climate in the first decades of the 22nd century depending on their response time (Fig. S1a). An analysis of the climate signal at the glaciers' mean altitude throughout the century reveals that air temperature, particularly in summer, is expected to be the main driver of glacier mass change in the region (Fig. 1). Interestingly, future warmer temperatures do not affect annual snowfall rates on glaciers as a result of both higher precipitation rates in the EURO-CORDEX ensemble (Fig. 1g–i)[26] and glaciers shrinking to higher elevations where precipitation rates are higher as a result of orographic precipitation enhancement[27]. The increase in glacier altitude also causes the solid to liquid precipitation ratio to remain relatively constant. Therefore, solid precipitation is projected to remain almost constant at the evolving glaciers' mean altitude independently from the future climate scenarios, while air temperature is projected to drive future glacier-wide mass changes (Fig. 1d, g). These results are in agreement with the main known drivers of glacier mass change in the French Alps[28]. Overall, the evolving glaciers are expected to undergo rather stable climate conditions under RCP 4.5, but increasingly higher temperatures and rainfall under RCP 8.5 (Fig. 1). These differences in the received climate signal are explained by the retreat of glaciers to higher altitudes, which keep up with the warming climate in RCP 4.5 but are outpaced by it under RCP 8.5.

The glacier ice volume in the French Alps at the beginning of the 21st century is unevenly distributed, with the Mont-Blanc massif accounting for about 60% of the total ice volume in the year 2015 (7.06 out of 11.64 km$^3$, Fig. 2a). The vast majority of glaciers in the French Alps are very small glaciers (<0.01 km$^2$), that are mainly remnants from the Little Ice Age, with a strong imbalance with the current climate[15]. Our projections highlight the almost complete disappearance of all glaciers outside the Mont-Blanc and Pelvoux

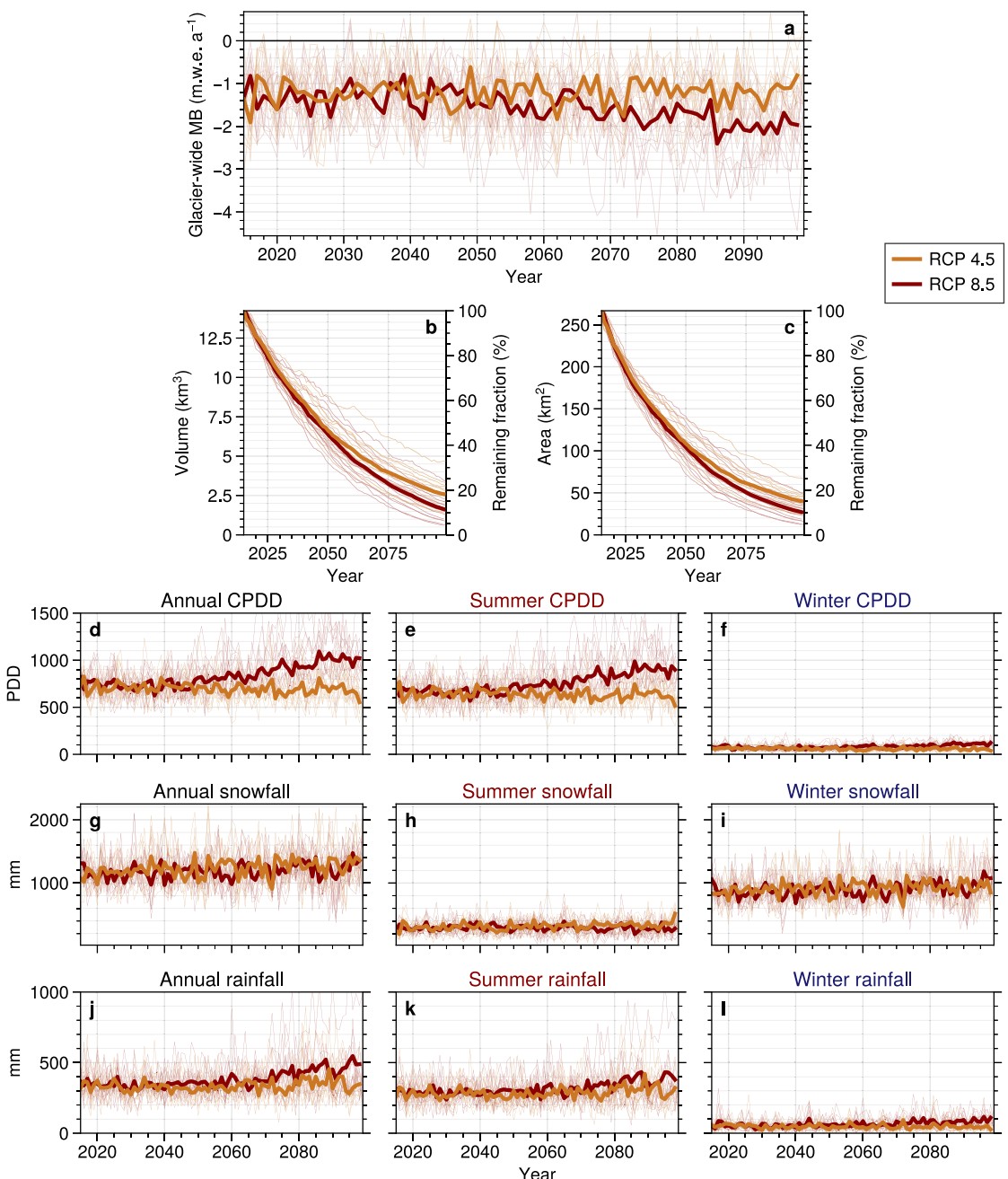

**Fig. 1 21st century glacier-wide MB, geometry and climate signal evolution of French Alpine glaciers. a** Glacier-wide annual MB, **b** Ice volume, **c** Glacier area. The cumulative positive degree days (CPDD), snowfall and rainfall **d–l**, are at the glaciers' annually evolving centroids. As such, these values reflect both the climatic forcing and the changing glacier geometry. Summer climate is computed between April 1st and September 30th and winter climate between October 1st and March 31st. Thin lines represent each of the 29 individual member runs, while the thick lines represent the average for a given RCP.

(Ecrins region) massifs under RCP 4.5 (Fig. 2) and RCP 8.5 by the end of the century. By 2100, under RCP 4.5, these two high-altitude massifs are predicted to retain on average 26% and 13% of their 2015 volume, respectively, with most of the ice concentrated in a few larger glaciers (>1 km², Fig. S4). Glacier landscapes are expected to see important changes throughout the French Alps, with the average glacier altitude becoming 300 m (RCP 4.5) and 400 m (RCP 8.5) higher than nowadays (Fig. 2a and S3).

**Nonlinear climate-glacier interactions.** Glacier surface mass changes are commonly modelled by relying on empirical linear relationships between PDDs and snow, firn or ice melt[8–10,29].

Since the climate and glacier systems are known to be nonlinear[13], we investigate the benefits of using a model treating, among others, PDDs in a nonlinear way in order to simulate annual glacier-wide MB at a regional scale. We compare model runs using a nonlinear deep learning MB model (the reference approach in our study) against a simplified linear machine learning MB model based on the Lasso[30], i.e. regularized multi-linear regression. Both MB models were trained with exactly the same data, and all other glacier model parameters were unchanged in order to allow isolating the effects of the non-linearities in the MB. A comparison between the two MB models shows that a nonlinear response to climate forcings is captured by the deep learning MB model, allowing a better representation of

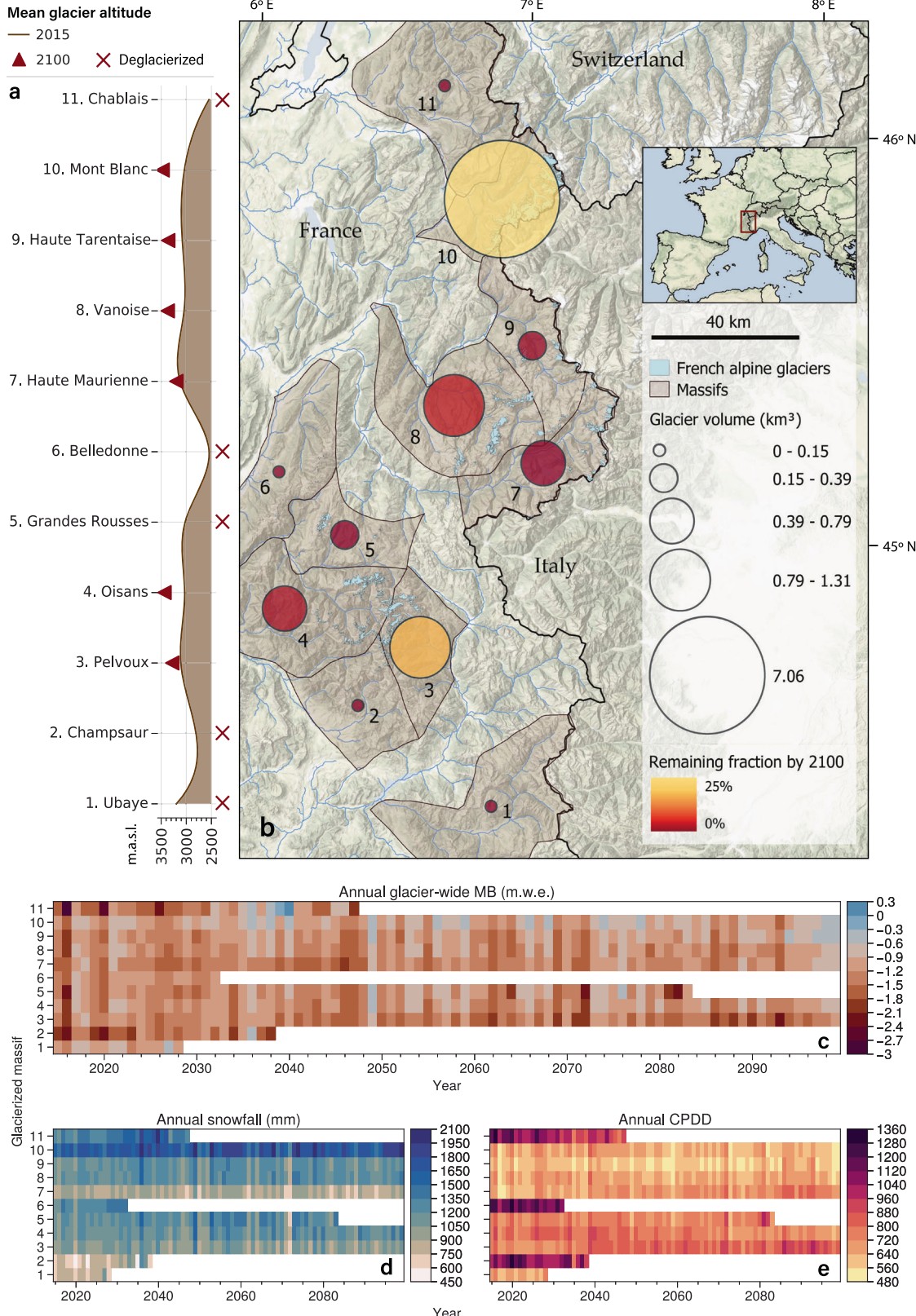

**Fig. 2 Projected glacier and climate evolution per glacierized massif between 2015 and 2100 under RCP 4.5. a** Projected mean glacier altitude evolution between 2015 and 2100. Massifs without glaciers by 2100 are marked with a cross, **b** Glacier ice volume distribution per massif, with its remaining fraction by 2100 (with respect to 2015), **c** Annual glacier-wide MB per massif, **d** Annual snowfall per massif, **e** Annual cumulative positive degree-days (CPDD) per massif. All values correspond to ensemble means under RCP 4.5. Years in white in c-e indicate the disappearance of all glaciers in a given massif.

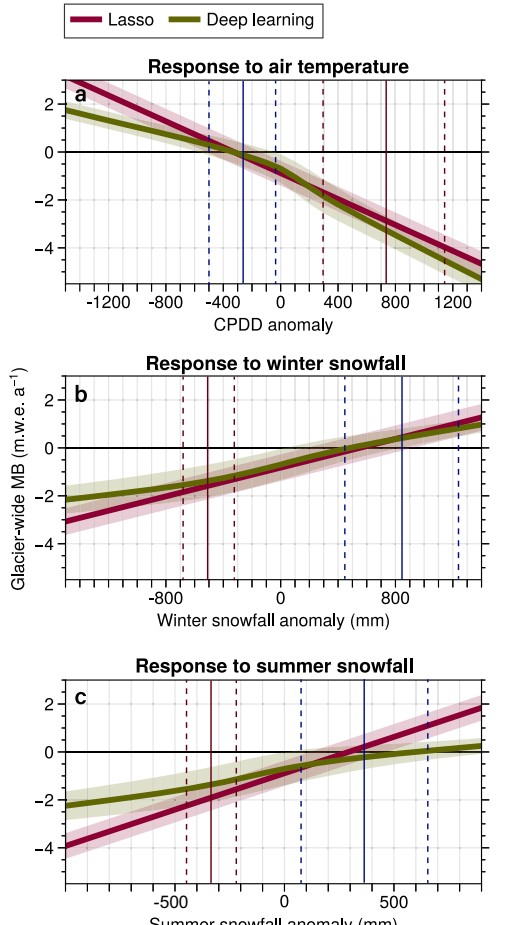

**Fig. 3 Nonlinear and linear response to climatic forcing of MB models.**
Nonlinear deep learning response and linear Lasso response to **a**
Cumulative positive degree days (CPDD) anomalies, **b** winter snowfall, and
**c** summer snowfall. All climate anomalies are computed with respect to the
1967–2015 mean values. Envelopes indicate ±σ based on results for all 660
glaciers in the French Alps for the 1967–2015 period. The vertical blue and
red lines indicate the distribution of extreme (top 5%) values for all 21st
century projected climate scenarios, with the mean value in the center and
±1σ indicated by dashed lines.

glacier mass changes, including significantly reduced biases for
extreme values (see Methods). A sensitivity analysis of both MB
models revealed nonlinear relationships between PDDs, snowfall
(in winter and summer) and glacier-wide MB, which the linear
model was only able to approximate ($r^2 = 0.41$ for the Lasso vs.
$r^2 = 0.76$ for deep learning in cross-validation[31]; Fig. 3).
Regarding air temperature forcings, the linear Lasso MB model
was found to be slightly under-sensitive to extreme positive
cumulative PDD (CPDD) and over-sensitive to extreme negative
CPDDs. Conversely, the linear MB model appears to be over-
sensitive to extreme positive and negative snowfall anomalies.
This behaviour is particularly clear for summer snowfall, for
which the differences are the largest (Fig. 3). In summary, the
linear approximations used by the Lasso manage to correctly fit
the main cluster of average values but perform poorly for extreme
values[31]. This has the strongest impact under RCP 2.6, where
positive MB rates are more frequent (Fig. 4), as the linear model
tends to over-estimate positive MB rates both from air tem-
perature and snowfall (Fig. 3). When using the linear MB model
(Lasso), glaciers are close to reaching an equilibrium with the
climate in the last decades of the century, which is not the case for

the nonlinear MB model (deep learning). Under warmer condi-
tions (RCP 8.5), the differences between the linear and nonlinear
MB model become smaller, as the topographical feedback from
glacier retreat compensates for an important fraction of the losses
induced by the late century warmer climate (Fig. S8 and Fig. 4).
This behaviour is expected for mountain glaciers, as they are
capable of retreating to higher altitudes, thus producing a positive
impact on their glacier-wide MB (Fig. 5). Alternatively, flatter
glaciers (*i.e.* ice caps) that are found in other glacierized regions
such as the Arctic, where the largest volumes of glacier ice (other
than the ice sheets) are stored[32], cannot retreat to higher eleva-
tions. This means that these flatter ice bodies, under a warming
climate, will be subject to higher temperatures than their steeper
counterparts. In order to investigate the implications of these
results for flat glaciers, we performed additional synthetic
experiments in order to reproduce this lack of topographical
feedback (Fig. 4). Our results point out that this lack of topo-
graphical feedback leads to an increased frequency of extreme
negative MB rates and to more pronounced differences between
the nonlinear and linear MB models (Figs. 4e and 5). Our pre-
vious work[31] has shown that linear MB models can be correctly
calibrated for data around the mean temperature and precipita-
tion values used during training, giving similar results and per-
formance to deep learning. However, as shown in our previous
work and confirmed here, the accuracy of linear models drasti-
cally drops as soon as the input climate data diverges from the
mean cluster of values used for training. A similar behaviour is
observed when comparing temperature-index models to more
complex models (e.g. energy balance), with differences increasing
when the conditions considerably differ from the calibration
period[33]. The lower fraction of variance explained by linear
models is present under all climate scenarios. Both models agree
around the average values seen during training (i.e. −0.78 m.w.e.
a⁻¹), but when conditions deviate from this mean training
data centroid, the Lasso can only linearly approximate the
extremes based on the linear trend set on the main cluster of
average values (Fig. 4). Overall, this results in linear MB models
overestimating both extreme positive (Fig. 4a, b) and negative
(Fig. 4e) MB rates.

We further assessed the effect of MB nonlinearities by
comparing our simulated glacier changes with those obtained
from other glacier evolution studies from the literature, which
rely on temperature-index models for MB modelling. Previous
studies on 21st century large-scale glacier evolution projections
have covered the French Alps[7,8]. Here, we compare our results
with those from a recent study that focused on the European
Alps[10]. In that study, a temperature-index model with a separate
degree-day factor (DDF) for snow and ice is used, resulting in
piecewise linear functions able to partially reproduce nonlinear
MB dynamics. Both the Lasso and the temperature-index
MB model rely on linear relationships between PDDs, solid
precipitation and MB. Therefore, their sensitivities to the
projected 21st century increase in PDDs are linear. Despite the
differences in the two modelling approaches (Table S2), both
regional glacier volume projections present relatively similar
results by the end of the century, with volume differences ranging
between 14% for RCP 2.6 to less than 2% for RCP 4.5 (Fig. S7).
Nonetheless, a close inspection of the annual glacier-wide MB
rates from both models reveals similar patterns to those found
when comparing deep learning and Lasso approaches (Figs. S5
and S6). Despite the existence of slightly different trends during
the first half of the century, both the Lasso and the temperature-
index model react similarly under RCP 4.5 and 8.5 during the
second half of the century, compared to the deep learning model.
The two models with linear MB responses to PDDs and
accumulation simulate more positive MB rates under RCP 2.6,

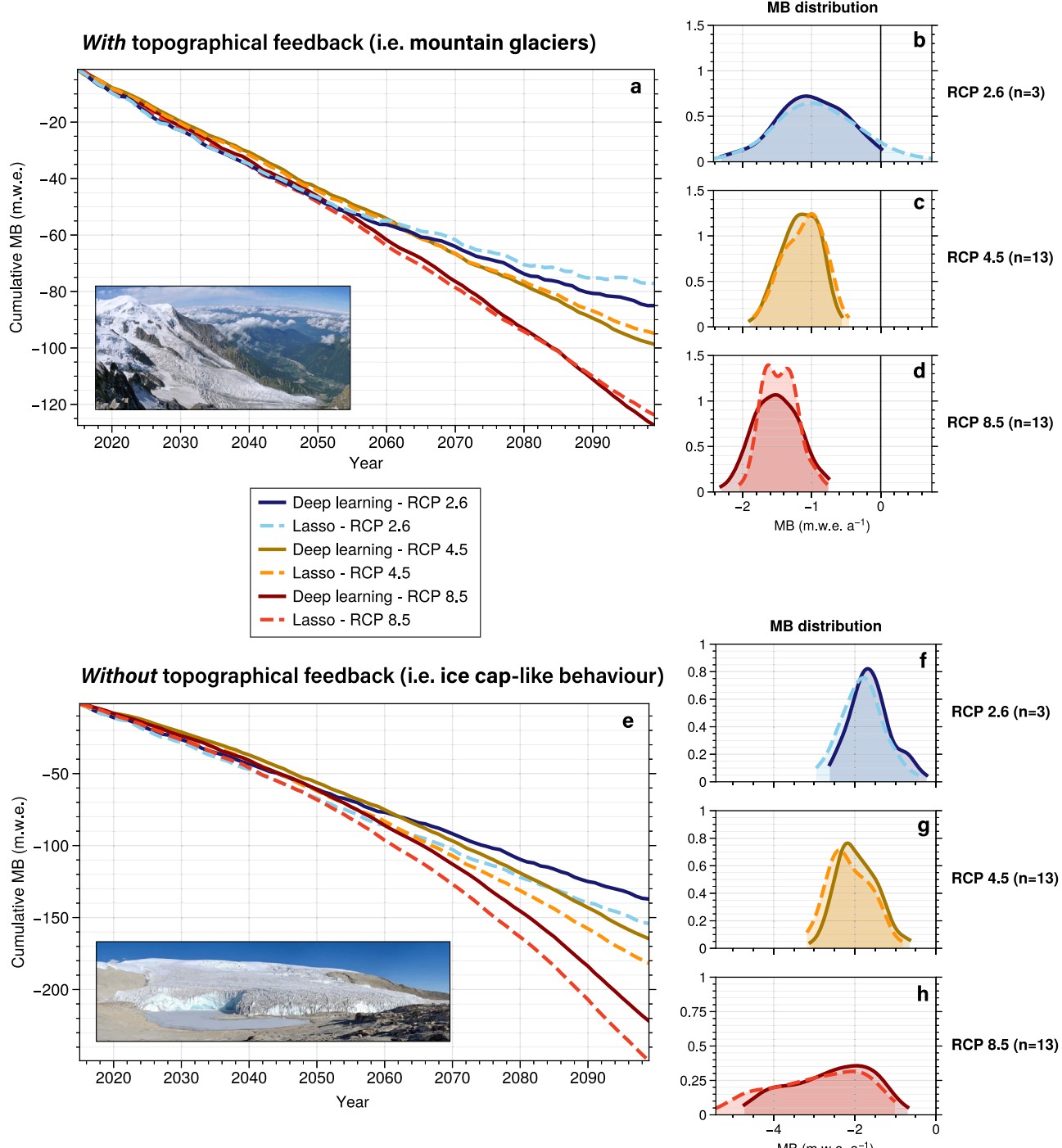

**Fig. 4 Effect of deep learning nonlinearities on glacier mass balance projections.** Average cumulative MB projections of French Alpine glaciers with a nonlinear deep learning *vs.* a linear Lasso model for 29 climate scenarios; **a** with topographical feedback (allowing for glacier retreat) and **e** without topographical feedback (synthetic experiment with constant mean glacier altitude). The projections without glacier geometry adjustment explore the behaviour of glaciers which cannot retreat to higher elevations (i.e. ice cap-like behaviour). **b**, **c**, **d** and **f**, **g**, **h** annual glacier-wide MB probability distribution functions for all *n* scenarios in each RCP. Vertical axes are different for the two analyses. Photographs taken by Simo Räsänen (Bossons glacier, European Alps, CC BY-SA 3.0) and Doug Hardy (Quelccaya ice cap, Andes, CC BY-SA 4.0).

highlighting their over-sensitivity to negative air temperature anomalies and positive snowfall anomalies (Fig. S6).

## Discussion
Studies have warned about the use of temperature-index models for snow and ice projections under climate change for

decades[34–36]. Temperature-index models are known to be over-sensitive to temperature changes, mainly due to important differences in the processes contributing to future warming. This oversensitivity directly results from the fact that temperature-index models rely on linear relationships between PDDs and melt and that these models are calibrated with past MB and climate data. In the past, shortwave radiation represented a more

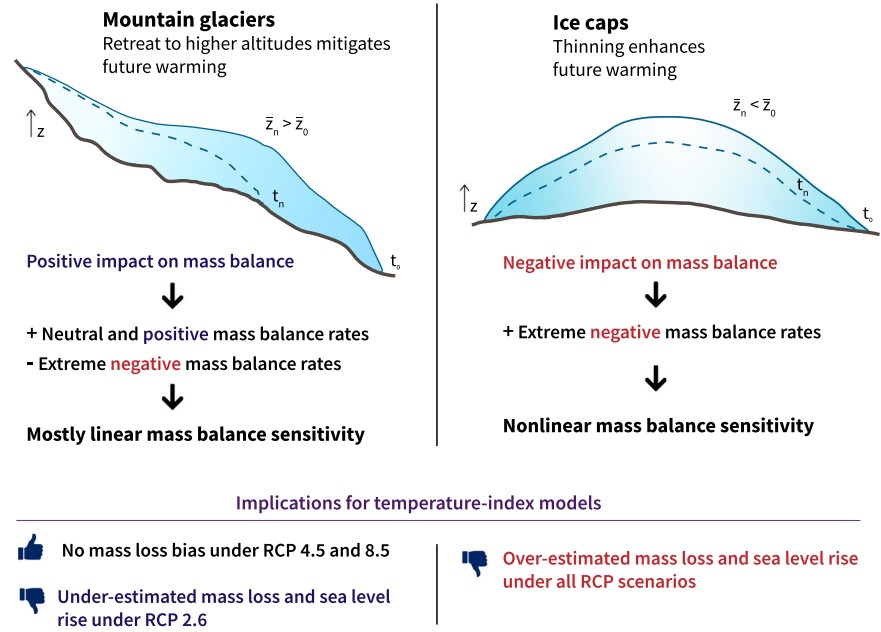

**Fig. 5 Main implications of nonlinear mass balance response to future warming and topographical feedback for mountain glacier and ice cap projections.** Glacier topography is a crucial driver of future glacier projections and is expected to play an important role in determining the magnitude that nonlinearities will have on the mass balance signal: ice caps and large flatter glaciers are expected to be more influenced by these nonlinear sensitivities than steep mountain glaciers in a warming climate. Graphics inspired by Hock and Huss[40].

important fraction in the glacier surface energy budget than the energy fluxes directly related to air temperature (e.g. longwave radiation budget, turbulent fluxes), in comparison with a future warmer climate. Indeed, the projected 21st century warming will lead to increasing incoming longwave radiation and turbulent fluxes, with no marked future trends in the evolution of short-wave radiation[37]. This will reduce the importance of shortwave radiation for future ablation rates, and it is expected to result in a reduction in values of degree-day factors (DDFs) and therefore a significant change in melt sensitivity to air temperature variations[36]. This behaviour has already been observed for the European Alps, with a reduction in DDFs for snow during the ablation season of $-7\%$ per decade[34]. Changes in DDFs with respect to air temperature also strongly depend on albedo, with ice presenting a substantially more nonlinear response than snow. Ice melt sensitivity to PDDs strongly decreases with increasing summer temperatures, whereas snow melt sensitivity changes at a smaller rate[34]. These trends explored with energy balance models from the literature correspond to the behaviour captured by our deep learning MB model, with a clearly less sensitive response of glacier-wide MB to extreme climate forcings, particularly in summer (Fig. 3c). This reduced sensitivity is captured through the response to summer snowfall anomalies, since the sensitivity to positive CPDD anomalies is quite similar for the linear and nonlinear models, as it encompasses both the accumulation and ablation seasons (Fig. 3a).

At this point, it is important to clarify the different ways of treating PDDs in the Lasso and the temperature-index MB models analysed in this study in order to justify analogies. The temperature-index model includes up to three different DDFs, for ice, firn and snow, resulting in three parameters. Alternatively, the Lasso model used here includes 13 DDFs: one for the annual CPDDs and 12 for each month of the hydrological year. Due to the statistical nature of the Lasso model, the response to snowfall anomalies is also highly influenced by variations in PDDs (Fig. 3c). This results in a higher complexity of the Lasso compared to a temperature-index model. Nonetheless, since they are

both linear, their calibrated parameters establishing the sensitivity of melt and glacier-wide MB to temperature variations remain constant over time. This is not the case for the nonlinear deep learning MB model, which captures the nonlinear response of melt and MB to increasing air temperatures, thus reducing the MB sensitivity to extreme positive and negative air temperature and summer snowfall anomalies (Fig. 3). Moreover, these differences between nonlinear and linear models appear to come from an over-sensitivity of linear models to increasing ablation season air temperatures, when ice is exposed in a large fraction of glaciers. Interestingly, this matches the nonlinear, less sensitive response to summer snowfall in the ablation season of our deep learning model (Fig. 3c), which is directly linked to summer air temperatures and has a strong influence on surface albedo. Conversely, during the accumulation season, glaciers are mostly covered by snow, with a much higher albedo and a reduced role of shortwave radiation in the MB that will persist even under climate change. This translates into a more linear response to air temperature changes compared to the ablation season (Fig. 3b). Despite their limitations, temperature-index models, owing to their simplicity and parsimonious data requirements, have been widely used for large-scale glacier projections[7,8]. Here, with our newly presented approach, we were able for the first time to quantify the effect that stationary parameters in temperature-index mass balance models have on transient glacier evolution. Our results serve as a strong reminder that the outcomes of existing large-scale glacier simulations should be interpreted with care, and that newly available techniques (such as the nonlinear deep learning approach presented here) and observations (e.g. on various mass balance and radiation components) are opening the door for updated and better constrained projections.

Our results also highlight the important role played by glacier geometry adjustment under changing climatic conditions, which is typical of mountain glaciers[38]. Our analysis suggests that due to this positive impact on the MB signal, only relevant differences are observed between nonlinear and linear MB models for the lowest emission climate scenarios (Fig. 4a). However, many

glacierized regions in the world present different topographical setups, with flatter glaciers, commonly referred to as ice caps, covering the underlying terrain[39]. Such ice caps cannot retreat to higher elevations in a warming climate, which inhibits this positive impact on MB[40] (Fig. 5). In fact, in many cases the surface lowering into warmer air causes this impact on the MB to be negative, further enhancing extreme negative mass balance rates. In order to investigate the effects of MB nonlinearities on flatter glaciers, we conducted a synthetic experiment using the French Alps dataset. We ran glacier evolution projections for both the deep learning and Lasso MB models, but we kept the glacier geometry constant, thus preserving the glacier centroid where the climate data is computed constant through time. With this setup, we reproduced the ice cap-like behaviour with a lack of topographical adjustment to higher elevations. The effect of glaciers shrinking to smaller extents is not captured by these synthetic experiments, but this effect is less important for flat glaciers that are dominated by thinning (Fig. 5). Additionally, glacier surface area was found to be a minor predictor in our MB models[31]. These synthetic experiments suggest that, for equal climatic conditions, flatter glaciers and ice caps will experience substantially more negative MB rates than steeper mountain glaciers. This translates into more frequent extreme negative MB rates, and therefore greater differences due to nonlinearities for the vast majority of future climate scenarios (Fig. 4e).

These different behaviours and resulting biases can potentially induce important consequences in long-term glacier evolution projections. On the one hand, MB nonlinearities for mountain glaciers appear to be only relevant for climate scenarios with a reduction in greenhouse gases emissions (Fig. 4a). The linear Lasso MB model suggests a stabilization of glacier evolution, reaching neutral MB rates by the end of the century. This behaviour is not observed with the nonlinear model, hinting at a positive bias of linear MB models under RCP 2.6. When comparing our deep learning simulations with those from the Lasso, we found average cumulative MB differences of up to 17% by the end of the century (Fig. S5b). For intermediate and pessimistic climate scenarios, no significant differences were found (Fig. S5c–f), except for the largest glaciers (e.g. Mer de Glace, 29 km$^2$ in 2015), which did show important differences under RCP 8.5 (up to 75%), due to their longer response time. This suggests that linear MB models are adequate tools for simulating MB of mountain glaciers with important topographical adjustment, with the only exception being the most optimistic climate scenarios and glaciers with long response times. On the other hand, for flatter glaciers large differences between deep learning and Lasso are obtained for almost all climate scenarios (Fig. 4e). Since these flatter glaciers are more likely to go through extreme negative MB rates, nonlinear responses to future warming play a more important role, producing cumulative MB differences of up to 20% by the end of the century (Fig. S5h, j, l). Therefore, linear MB models present more limitations for projections of ice caps, showing a tendency to negative MB biases. As previously mentioned, here these differences are computed at regional level for a wide variety of glaciers. Differences for individual glaciers can be much more pronounced, as large and flat glaciers will have topoclimatic configurations that produce more extreme MB rates than small and steep glaciers with a short response time.

The main uncertainties in future glacier estimates stem from future climate projections and levels of greenhouse gas emissions (differences between RCPs, GCMs, and RCMs), whose relative importance progressively increases throughout the 21st century. With a secondary role, glacier model uncertainty decreases over time, but it represents the greatest source of uncertainty until the middle of the century[8]. Taking into account that for several regions in the world about half of the glacierized volume will be lost during this first half of the 21st century, glacier models play a major role in the correct assessment of future glacier evolution. The two recent iterations of the Glacier Model Intercomparison Project (GlacierMIP[7,8]) have proved a remarkable effort to aggregate, compare and understand global glacier evolution estimates and their associated uncertainties. Despite the existence of a wide variety of different approaches to simulate glacier dynamics, all glacier models in GlacierMIP rely on MB models with linear relationships between PDDs and melt, and precipitation and accumulation. Some of these models use a single DDF, while others have separate DDFs for snow and ice, producing a piecewise function composed of two linear sub-functions that can partially account for nonlinear MB dynamics depending on the snowpack. As we have previously shown, these models present a very similar behaviour to the linear statistical MB model from this study (Fig. 4 vs. S5). Interestingly, our analysis indicates that more complex models using separate DDFs for ice, firn and snow might introduce stronger biases than more simple models using a single DDF. DDFs are known to vary much less with increasing temperatures for intermediate values of albedo (i.e. 0.5) than lower values typical from ice[34]. Consequently, a simple MB model with a single DDF (e.g. the Open Global Glacier Model - OGGM[9]) is likely to be less affected by an over-sensitivity to future warming than a more complex model with dedicated DDFs for ice, snow, and firn. This creates an interesting dilemma, with more complex temperature-index MB models generally outperforming simpler models for more climatically homogeneous past periods but introducing important biases for future projections under climate change. Our results suggest that, except for the lowest emissions climate scenarios and for large glaciers with long response times, MB models with linear relationships for PDDs and precipitation are suitable for mountain glaciers with a marked topographical feedback. On the one hand, this improves our confidence in long-term MB projections for steep glaciers made by most GlacierMIP models for intermediate and high emissions climate scenarios. On the other hand, ice caps present a different response to future warming, with our results suggesting a negative MB bias by models using linear PDD and accumulation relationships. Ice caps in the Canadian Arctic, the Russian Arctic, Svalbard, and parts of the periphery of Greenland are major reservoirs of ice, as well as some of the biggest expected contributors to sea level rise outside the two polar ice sheets[7]. On top of that, they happen to be among the glacierized regions with the largest projected uncertainties[8]. Together with recent findings by another study[41] highlighting the increased uncertainties in ice thickness distribution estimates of ice caps compared to mountain glaciers, our results raise further awareness on the important uncertainties in glacier projections for ice caps. These conclusions drawn from these synthetic experiments could have large implications given the important sea-level contribution from ice cap-like ice bodies[8]. However, to further investigate these findings, experiments designed more towards ice caps, and including crucial mechanisms such as ice-ocean interactions and thermodynamics, should be used for this purpose.

In this study, we demonstrated the advantages of using deep learning to model glacier MB at regional scales, both in terms of variance and bias. Nonetheless, a better understanding of the underlying processes guiding these nonlinear behaviours at large geographical scales is needed. The machine learning models used in this study are useful to highlight and quantify how nonlinearities in MB affect climate-glacier interactions, but are limited in terms of process understanding. At present, using complex surface energy balance models for large-scale glacier projections is not feasible yet, mainly due to the lack of input data. Therefore, an alternative nonlinear parameterization for the reduction in MB sensitivity under increasing air temperatures would be useful.

This is particularly important for the ablation season and for ice DDFs, which need to accommodate the progressively decreasing role that shortwave radiation will play in the future glacier surface energy budget under warmer conditions. New methods bridging the gap between domain-specific equations and machine learning are starting to arise[42], which will play a crucial role in further investigating the physical processes driving these nonlinear climate-glacier interactions.

By unravelling nonlinear relationships between climate and glacier MB, we have demonstrated the limitations of linear statistical MB models to represent extreme MB rates in long-term projections. Our analyses suggest that these limitations can also be translated to temperature-index MB models, as they share linear relationships between PDDs and melt, as well as precipitation and accumulation. By performing glacier projections both with mountain glaciers in the French Alps and a synthetic experiment reproducing ice cap-like behaviour, we argue that the limitations identified here for linear models will also have implications for many other glacierized regions in the world. Uncertainties of existing projections of future glacier evolution are particularly large for the second half of the 21st century due to a large uncertainty on future climatic conditions. Our results indicate that these uncertainties might be even larger than we previously thought, as linear MB models are introducing additional biases under the extreme climatic conditions of the late 21st and 22nd centuries. Through synthetic experiments, we showed that the associated uncertainties are likely to be even more pronounced for ice caps, which host the largest reserves of ice outside the two main ice sheets[32]. This implies that current global glacier mass loss projections are too low for the lowest emissions climate scenarios and too high for the highest emissions ones, which has direct consequences for related sea-level rise and water resources projections.

## Methods

**Glacier mass balance modelling.** Glacier-wide MB is simulated annually for individual glaciers using deep learning (i.e. a deep artificial neural network) or the Lasso (regularized multilinear regression)[30]. This modelling approach was described in detail in a previous publication dedicated to the methods, where the ALpine Parameterized Glacier Model (ALPGM[43]) was presented[31]. ALPGM uses a feedforward fully connected multilayer perceptron, with an architecture (40-20-10-5-1) with Leaky-ReLu[44] activation functions and a single linear function at the output. A He uniform initialization[45] was used for the network parameters. The smallest best performing architecture was used, in order to find a good balance between predictive power, speed, and extrapolation outside the training data. The training was performed with an RMSprop optimizer, batch normalization[46], and we used both dropout and Gaussian noise in order to regularize it. The Lasso[30], used for the linear mass balance model, is a linear regression analysis method which shrinks model parameters, thus performing both variable selection and regularization. A dataset of 32 glaciers with direct annual glacier-wide MB observations and remote sensing estimates was used to train the models. For these 32 glaciers, a total of 1048 annual glacier-wide MB values are available, covering the 1967–2015 period with gaps. In order to simulate annual glacier-wide MB values, (a) topographical and (b) climate data for those glaciers and years were compiled for each of the 1048 glacier-year values. (a) Topographical predictors were computed based on the glaciers' annually updated digital elevation model (DEM). These predictors are composed of: the mean glacier altitude, maximum glacier altitude, slope of the lowermost 20% altitudinal range of the glacier, glacier surface area, latitude, longitude and aspect. (b) Climate predictors are based on climatic anomalies computed at the glaciers' mean altitude with respect to the 1967–2015 reference period mean values. Models were trained using the SAFRAN reanalysis dataset[47], including observations of mountain regions in France for the 1958–2015 period. This reanalysis is specifically designed to represent meteorological conditions over complex mountain terrain, being divided by mountain massif, aspect and elevation bands of 300 m. Winter climate data are computed between October 1 and March 31, and summer data between April 1 and September 30. Climate predictors consist of: the annual CPDD, winter snowfall, summer snowfall, monthly temperature and monthly snowfall. This creates a total of 34 input predictors for each year (7 topographical, 3 seasonal climate, and 24 monthly climate predictors).

In order to avoid overfitting, MB models were thoroughly cross-validated using all data for the 1967–2015 period in order to ensure a correct out-of-sample performance. Three different types of cross validation were performed: a Leave-One-Glacier-Out (LOGO), a Leave-One-Year-Out (LOYO) and a Leave-Some-Years-and-Glaciers-Out (LSYGO). Each one of these cross-validations served to evaluate the model performance for the spatial, temporal and both dimensions, respectively. When working with spatiotemporal data, it is imperative to respect spatial and temporal data structures during cross-validation in order to correctly assess an accurate model performance[48]. With this cross-validation we determined a deep learning MB model spatiotemporal (LSYGO) RMSE of 0.59 m.w.e. a$^{-1}$ and a r$^2$ of 0.69, explaining 69% of the total MB variance. Alternatively, the Lasso MB model displayed an RMSE of 0.85 m.w.e. a$^{-1}$ and an r$^2$ of 0.35[31]. Simulations for projections in this study were made by generating an ensemble of 60 cross-validated models based on LSYGO. Each one of these models was created by training a deep learning model with the full dataset except all data from a random glacier and year, and evaluating the performance on these hidden values. This ensures that the model is capable of reproducing MB rates for unseen glaciers and years. Simulations were then performed by averaging the outputs of each one of the 60 ensemble members. This approach is known as a cross-validation ensemble[49]. Future projections of glacier-wide MB evolution were performed using climate projections from ADAMONT[25]. This dataset applies a statistical adjustment specific to French mountain regions based on the SAFRAN dataset, to EURO-CORDEX[26] GCM-RCM-RCP members, covering a total of 29 different future climate scenarios for the 2005–2100 period. This adjustment represents a major improvement over most climate data used to force regional and global glacier models. The high spatial resolution enables a detailed representation of mountain weather patterns, which are often undermined by coarser resolution climate datasets.

**Glacier geometry evolution.** A well-established parametrization based on empirical functions[50] was used in order to redistribute the annually simulated glacier-wide mass changes over each glacier. This parametrization reproduces in an empirical manner the changes in glacier geometry due to the combined effects of ice dynamics and MB. As for the MB modelling approach, a detailed explanation on this method can be found in a previous dedicated paper on the methods[31]. In our model, we specifically computed this parameterized function for each individual glacier larger than 0.5 km$^2$, representing 80% of the total glacierized area in 2015, using two DEMs covering the whole French Alps: a photogrammetric one in 1979 and a SPOT-5 one in 2011. We previously demonstrated that this period is long enough to represent the secular trend of glacier dynamics in the region[31]. Both DEMs were resampled and aligned at a common spatial resolution of 25 m. For each glacier, an individual parameterized function was computed representing the differences in glacier surface elevation with respect to the glacier's altitude within the 1979–2011 period. This method has the advantage of including glacier-specific dynamics in the model, encompassing a wide range of different glacier behaviours. Using this function, the glacier-specific ice thickness and the DEM are updated every year, adjusting the 3D geometry of each glacier. This enables the recalculation of every topographical predictor used for the MB model, thus updating the mean glacier altitude at which climate data for each glacier are retrieved. This annual geometry adjustment accounts for the effects of glacier retreat on the climate signal received by glaciers. Glaciers smaller than 0.5 km$^2$ often display a high climate imbalance, with their equilibrium line being higher than the glacier's maximum altitude. Such glaciers are often remnants of the Little Ice Age, and mainly lose mass via non-dynamic downwasting[51]. For such cases, we assumed that ice dynamics no longer play an important role, and the mass changes were applied equally throughout the glacier.

The performance of this parametrization was validated in a previous study, indicating a correct agreement with observations[31]. The dataset of initial glacier ice thickness, available for the year 2003, determines the starting point of our simulations. We performed a validation simulation for the 2003–2015 period by running our model through this period and comparing the simulated glacier surface area of each of the 32 glaciers with MB to observations from the 2015 glacier inventory[16,52]. Then, we ran multiple simulations for this same period by altering the initial ice thickness by ±30% and the glacier geometry update parametrizations by ±10%, according to the estimated uncertainties of each of the two methods[31]. These results revealed that the main uncertainties on glacier simulations arise from the initial ice thickness used to initialize the model. This is well in agreement with the known uncertainties of glacier evolution models, with glacier ice thickness being the second largest uncertainty after the future GCM-RCM-RCP climate members used to force the model[29]. Glacier ice thickness observations are available for four different glaciers in the regions, which are compared to the estimates used in this model. Ice thickness accuracy varied significantly, with an overall correct representation of the ice distribution but with local biases reaching up to 100%. The ice thickness data for two of the largest glaciers in the French Alps were modified in order to improve data quality. Ice thickness data for Argentière glacier (12.27 km$^2$ in 2015) was taken from a combination of field observations (seismic, ground-penetrating radar or hot-water drilling[53]) and simulations[32]. The estimated ice thickness for Mer de Glace (28.87 km$^2$ in 2015) was increased by 25% in order to correct the bias with respect to field observations[31]. Since these two glaciers are expected to be some of the few large glaciers that will survive the 21st century climate, an accurate representation of their initial ice thickness has an important effect on the estimates of remaining ice.

**Model comparison and extraction of nonlinearities.** The nonlinearities present in the simulated annual glacier-wide MB values were assessed by running two different glacier simulations with two different MB models. The advantage of this

method is that by only changing the MB model, we can keep the rest of the model components (glacier dynamics and climate forcing) and parameters the same in order to have a controlled environment for our experiment. Therefore, we were capable of isolating the different behaviours of the nonlinear deep learning model and a linear machine learning model based on the Lasso[30]. Both machine learning MB models were trained with exactly the same data coming from the 1048 annual glacier-wide MB values, and both were cross-validated using LSYGO. In order to investigate the effects of MB nonlinearities on ice caps, we performed the same type of comparison between simulations, but the glacier geometry update module described in the "Glacier geometry evolution" section was deactivated. This synthetic setup allowed us to reproduce the climatic conditions to be undergone by most ice caps, with their mean surface altitude hardly evolving through time. This removes the topographical feedback typical from mountain glaciers, and reproduces the more extreme climate conditions that ice caps are likely to endure through the 21st century[40]. This synthetic experiment is an approximation of what might occur in other glacierized regions with ice caps. In many aspects, it might be too optimistic, as many ice caps will have a negative impact on MB through thinning, bringing their mean surface elevation to lower altitudes, thus further warming their perceived climate. This means that these differences linked to MB nonlinearities observed in this experiment could be even greater for such ice caps. However, the impact of different climate configurations, such as a more continental and drier climate or a more oceanic and humid climate, would certainly have an impact on the results, albeit a much less important one than the lack of topographical feedback explored here. Our synthetic experiment does not account for glacier surface area shrinking either, which might have an impact on the glacier-wide MB signal. Nevertheless, we previously demonstrated that glacier surface area is not an important predictor of MB changes in our models[29], and ice caps evolve mostly through thinning and not shrinking (Fig. 5).

Additionally, the specific responses of the deep learning and Lasso MB models to air temperature and snowfall were extracted by performing a model sensitivity analysis. Since the neural network used here virtually behaves like a black box, an alternative way is needed to understand the model's behaviour. In order to do so, we applied a deterministic sampling process as a sensitivity analysis to both the deep learning and the Lasso MB models. For that, a dataset of input predictors covering all the glaciers in the French Alps for the 1967–2015 period was generated from a past MB reconstruction study[15]. Multiple copies of this dataset were created, and for each individual copy a single predictor (i.e. CPDD, winter snowfall or summer snowfall) was modified for all glaciers and years. This allows us to assess the MB models responses at a regional scale to changes in individual predictors (Fig. 3). Regarding air temperature, a specific CPDD anomaly ranging from −1500 PDD to +1500 PDD in steps of 100 PDD was prescribed to all glaciers for each dataset copy. Since both MB models also include monthly temperature data as predictors, this CPDD anomaly was distributed evenly between the ablation season (April 1–September 30), following the expected increase in mostly summer temperatures instead of winter temperatures in the future (Fig. 1). Tests were performed distributing the CPDD anomalies equally among all months of the year with very similar results. The same was done with winter snowfall anomalies, ranging between −1500 mm and +1500 mm in steps of 100 mm, and summer snowfall anomalies, ranging between −1000 mm and +1000 mm in steps of 100 mm. The anomaly in snowfall was evenly distributed for every month in the accumulation (October 1–April 31) and ablation seasons, respectively. This experiment enabled the exploration of the response to specific climate forcings of a wide range of glaciers of different topographical characteristics in a wide range of different climatic setups, determined by all meteorological conditions from the years 1967–2015 (Fig. 3).

Alternatively, the comparisons against an independent large-scale glacier evolution model were less straightforward to achieve. GloGEMflow[10] is a state-of-the-art global glacier evolution model used in a wide range of studies, including the second phase of GlacierMIP[7,8]. Several differences are present between ALPGM, the model used in this study, and GloGEMflow (Table S2), which hinder a direct meaningful comparison between both. In order to overcome these differences, some adaptations were performed to the GloGEMflow output, accompanied with some hypotheses to ensure a realistic comparison. The first main difference is related to the climate data used to force the models. GloGEMflow relies on EURO-CORDEX ensembles[26], whereas ALPGM uses ADAMONT[25], an adjusted version of EURO-CORDEX specifically designed for mountain regions. ADAMONT provides climate data at 300 m altitudinal bands and different slope aspects, thus having a significantly higher spatial resolution than the 0.11° from EURO-CORDEX. This implies that specific climatic differences between massifs can be better captured by ALPGM than GloGEMflow. Nonetheless, since the main GCM-RCM climate signal is the same, the main large-scale long-term trends are quite similar. We reduced these differences by running simulations with GloGEMflow using exactly the same 29 climate members used by ALPGM in this study (Table S1). The initial glacier ice thickness data for the year 2003 also differs slightly between both models. The original ice thickness estimates of the methods used by both models are different[10,32], and for ALPGM we performed some additional modifications to the two largest glaciers in the French Alps (see 'Glacier geometry evolution' for details). Despite these differences, the average altitude difference of the glaciers between both models is never greater than 50 m (Fig. S10). Since in ALPGM the climate forcing of glaciers is extracted at the mean glacier altitude, we do not expect these altitude differences to drive important MB

differences between models. Another source of discrepancy between both models comes from the different MB data used to calibrate or train the MB models. GloGEMflow has been previously applied in a study over the whole European Alps, and its temperature-index model was mainly calibrated with MB data from the Swiss Alps. Swiss glaciers have displayed less negative MB rates than French glaciers during the last decades, thus likely introducing a bias in simulations specific to the French Alps. In order to improve the comparability between both models, a MB bias correction was applied to GloGEMflow's simulated MB, based on the average annual MB difference between both models for the 2003–2015 period (−0.4 m.w.e. a$^{-1}$) over the French Alps. Finally, there are differences as well in the glacier dynamics of both models, with ALPGM using a glacier-specific parameterized approach and GloGEMflow explicitly reproducing the ice flow dynamics. Nonetheless, these differences have been shown to be rather small, having a lower impact on results than climate forcings or the initial glacier ice thickness[10].

## Data availability
The model output data generated in this study have been deposited in netCDF and CSV format in a Zenodo repository under accession code Creative Commons Attribution 4.0 International. https://zenodo.org/record/5549758.

## Code availability
The source code of the glacier model can be freely accessed in the following repository: https://github.com/JordiBolibar/ALPGM.

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

## Acknowledgements

We acknowledge the more than 50 years of glaciological monitoring performed by the GLACIOCLIM French National Observatory (https://glacioclim.osug.fr), which provided essential observations for our modelling study. This work was funded by the Labex OSUG@2020 (Investissements d'avenir, ANR10 LABX56) and the Auvergne-Rhône-Alpes region through the BERGER project. J.B. was supported by a NWO VIDI grant 016.Vidi.171.063. H.Z. acknowledges the funding received from a EU Horizon 2020 Marie Skłodowska-Curie Individual Fellowship (grant no. 799904) and from the Fonds de la Recherche Scientifique – FNRS (postdoctoral grant – chargé de recherches).

## Author contributions

J.B. developed the main glacier model, performed the simulations, analysed the results, and wrote the paper. A.R. provided glacier mass balance data and performed the glaciological analyses. I.G. contributed to the climate analyses. H.Z. performed simulations with another glacier model, provided results for comparison, and contributed to the glaciological analyses. C.G. contributed to the extraction of nonlinear mass balance responses and to the statistical analysis. All authors provided inputs to the paper and helped to write it.

## Competing interests

The authors declare no competing interests.
