## [Peer Review File · Nature Communications]

Nonlinear sensitivity of glacier mass balance to future climate change unveiled by deep learningReviewers' Comments:

Reviewer #1:

Remarks to the Author:

Dear Authors

The manuscript titled "Deep learning unveils non-linear climate-glacier interactions through the 21st-century deglaciation of the French Alps" is exciting. The manuscript is an excellent contribution towards
Glaciology/climatology.

I have some comments to improve the quality of the manuscript.

1. Page 12, Line 206: Authors refer to models from previous publications; however, it will be significant if authors can provide information on model architecture, training information, parameter optimization, etc.
2. Have authors tried Shallow learning or machine learning models? Authors should add information about recent shallow neural networks/ML/DL applied in glaciology.
3. The word Deep Learning seems to be overused; instead, the model name should be given because Deep Learning itself is an extensive area comprising LSTM, CNN, hybrid models, etc.
4. The methodology section is not providing the core of models, which is a weaker part of this manuscript.
5. Page 14, Line 393: What is how-water drilling?
6. Page 16 Line 467-472: "A linear regression model....ice volume in 2015." How can a linear model provide reliable estimates for such a non-linear behavior of glacier ice volume forecasting? Have the authors tried the sensitivity analysis of this regression?

The manuscript requires some brief yet essential information on the model characteristics, as discussed in earlier comments.

Reviewer #2:

Remarks to the Author:

General comments

This manuscript uses a previously-established deep learning approach of simulating glacier change (mass balance and ice dynamics). I am not an expert in deep learning or machine learning, but I see that the method is already published in Bolibar et al., 2020, so hopefully has already been assessed by experts. Here, the deep learning method is applied to simulating glacier change in the French Alps through the 21st century. These future projections using the deep learning method are then compared to results from a linear machine learning model to better understand non-linear components of the glacier-climate system. This component of the paper- investigating the impact of linear versus non-linear models on glacier mass balance, is interesting and an important contribution to glaciology. As the authors note, the majority of future glacier projections currently use temperature index models, so better-understanding weaknesses or biases in these models is important. The paper is well written, especially the introduction, and nicely ties in the big picture/larger importance of this work in the main text and especially in the Supplementary Information text.

In the paper, linear mass balance models are associated with temperature-index models (e.g. ling

28). However, the linear mass balance model that is compared with the deep learning approach is another machine learning model (LASSO). A few suggestions around this point include: 1) It should be clear that the results are being compared with a machine learning approach and not a traditional temperature-index model. 2) I would have liked to see the results of the deep learning approach also compared with the temperature-index model, which most glaciologists use and or would understand. Right now, results are compared with results of GloGEMflow, but with the only analysis being Fig S5. 3) I also wish there was a brief description of the LASSO model in this paper when it is first introduced, right now the authors reference a previous paper.

A main finding of the paper is that deep learning captures important nonlinearities in the glacier/climate system. The authors nicely investigate this concept and present the results. It is interesting that the impact of the different approaches on cumulative mass balance is very minor (Fig 4j). I think it needs to be clear that while the deep learning approach does improve the simulation of extreme positive and negative mass balances, the results here don't show a huge difference on cumulative mass balance. A discussion point could be added that centers around the most appropriate problems to apply a deep learning approach (when capturing extreme extreme glacier mass balance years is important) versus when a traditional temperature-index model is sufficient (when capturing cumulative glacier mass balance is the goal, at least suggested by these results).

I would have liked more discussion on the description and significance of Figure 4. This is the main Figure showing the importance of the deep learning approach to capturing nonlinearities, which I think is the biggest contribution of this paper.

Specific comments

L62 - "strongest glacier retreat" not clear what strongest means, fastest?

Fig 1 - Why not have Figures S1 & S2 here? The supplementary figures are useful in showing all 3 RCP scenarios, which would be nice to include in the main text.

L159 - not clear what the correlations are for, modeled vs measured glaciological mass balance? Modeled vs measured PDDs and snowfall (from what climate data)?

Fig 3 - Would also be interesting to see climate anomalies as a %. For example, -1200 or +1200 CPDD anomalies seem like huge differences from normal. The extreme values are where the two models diverge, is there a way to show how often anomalies fall in those extreme values.

Figure 4c,f,i - does the deep learning model never result in positive mass balance? Figures a,d,g show there are years with positive mass balance. Why are the PDFs all negative?

L297 - 'remarkable' is a bit opinionated, is there a better word?

Lauren Vargo
Research Fellow
Antarctic Research Centre
Victoria University of Wellington

Reviewer #3:

Remarks to the Author:

This paper applies a data science method of modeling glacier mass adjustment to climate in the

French Alps for the coming century when they drive the model with an ensemble of multiple future models under three emission scenarios (RCP 2.6, 4.5, 8.5). The extent and nature of the predicted glacier mass balance (MB) response is alarming: by end of this century, fully 75 to 88% of the glacier volume will be lost, and what remains will be dramatically altered landscapes. It is really intriguing to see the disappearance of mean surface area, emphasizing loss of the smaller and lower glaciers. It is also eye-opening to see that cumulative MB differences were mitigated by a complete loss of half the glaciers by mid-century. The unique contribution the authors claim is in the methodology: that this deep learning approach improves upon linear mass balance modeling.

The authors champion their neural network (deep learning) method as being an important new contribution for the scientific community, for which they provide a more complete cross-validation in a prior paper using the same dataset. They demonstrate that deep learning does give "superior nonlinear explained variance," making models more accurate because both climate and glacier systems are known to react non-linearly to forcing. This is especially important as glaciers and climate deviate more extremely from mean conditions that appear most often in calibration. In this paper, they focus on the results of running model with 29 different climate members. They also provide a test of this relative mass balance response to linear vs nonlinear idea by comparing linear regression model to non-linear deep learning (artificial neural network) model of MB. They use the same predictors that are both topographic and climatological. They show that for more extreme values, the non-linear model makes a profound difference. However, the problem here is that for most of the 21st century, there is very little difference in the models; linear and nonlinear MB simulations are nearly identical.

In assessing the merit and impact of this work, one wonders about the appropriateness of this national focus on just a small subset of glaciers in France only. Are these new insights of sufficient scope? What new insights do we gain about the future of Alps? And what about other regions where the authors suggest such a modeling approach would be more important? Does this method improve what would be predicted by a more temperature dependent index method that the authors categorize under the generalized 'linear' methods? Furthermore, what are the actual physical process insights elucidated by this approach? As a data-driven method, it is not solving fundamental equations of dynamics.

I'm persuaded that this is a novel method applied to glacier change studies, but I lack adequate knowledge to give thorough technical review of the deep learning. I think that a more compelling case could be made to justify this data-analytical method by showing how it better captures physical processes, or allows for specific new predictive capacity for water management in the future.

I think more discussion about the specific 'non-linearities' and their implications would help. The non-linear deep learning is not obvious to the non-expert. As written, the authors show how mass balance responds non-linearly to some specific variables in Fig 3, but don't explain why in terms of physical processes. It is insufficient to claim that climate and glacier systems are known to be nonlinear (L149). What does it mean to treat PDD in a non-linear way? Are there feedbacks or sensitivities of mass balance to specific variables that are characterized by nonlinear models? If so, how? Furthermore, are they really that big of a deal when it seems that we won't see much difference for 50 more years, and by then there will be so much less ice anyway that it is perhaps "much ado about nothing"? As written, the paper suggests that uncertainties "might be even greater than we previously thought" for future climate scenarios, but does not hypothesize why or where. The authors claim that their work is very similar to the linear models, and only diverge in extremes. They speculate that the impacts could be more pronounced in complex regions. However, this is hard to grasp, since the method is effectively as good as the data.

It is interesting to see how initial ice thickness is the main source of uncertainties of glacier simulations. Only 4 of the glaciers have such data. Is there a need to get more of these measurements?

Some line edits:

L23: The abstract mentions 75 and 88% glacier volume loss by end of century. Loss from what point, the 2021 currently existing volume? Explain/clarify.

L50: delete the 'in order' before to

L53: type of model

L103-106: The description of the future snowfall rates on glaciers not being effected is not clear as written. Maybe articulating more clearly about elevation dependent ratio of solid to liquid precipitation would make this more evident.

L137: would be better worded as, "...Little Ice Age that are strongly out of balance with the current climate."

L138: change wording; our projections describe the almost COMPLETE disappearance

L140: instead of store, I recommend "retain"

L141: large should be "larger" and give a dimension to match the Fig S4 (>2 km²)

L187: rely should be "relies"

L228 : change are to "comprise"

L242: typo? Not sure what is Fig. "4Sb"

L246: also, 5Se

L268-270: this seems a stretch beyond the data to speculate that heterogeneous topography and "climatic char's" are likely to go thru higher variety of climate extremes. On what basis?

L274: sentence ends incompletely.

L275-278: awkward word choice/prose: "in a first term" and "in a second term." Do the authors intend this as an idiom like, "on one hand," or is it referring to an actual quantitative measure of uncertainty term? Please clarify.

L354: should be combined effects

Fig. 2: (a) confusing shading below the 2015 mean elevation; makes it seem like topography rather than median elevation.

In c - e panels, lines all have end points; is that when respective glaciers disappear? Should be mentioned/clarified.

Also, there appear to be random pixelated yrs of positive MB in (c). Why?

Caption to Fig. 2 (c) has a spurious "A"

Fig. 4: the RCP 8.5 graphs are strange. Annual nonlinear difs starts in positive values. How? And the cumulative nonlinear difference goes positive. Is that right? Explain.

Table S1: climate 'members' to force glacier evolution model

Bolibar et al. responses to editor and reviewer's comments

Author responses are given *in italic* after each of the editor and reviewer's comments.

Editor

Thank you again for submitting your manuscript "Deep learning unveils nonlinear climate-glacier interactions through the 21st century deglaciation of the French Alps" to Nature Communications. We have now received reports from 3 reviewers and, after careful consideration, we have decided to invite a major revision of the manuscript.

As you will see from the reports copied below, the reviewers raise important concerns. We find that these concerns limit the strength of the study, and therefore we ask you to address them with additional work. Without substantial revisions, we will be unlikely to send the paper back to review. In particular, reviewers note instances where additional analyses would support the robustness and broad applicability of the methods and conclusions. Reviewer #1 highlights where more detail on the model and sensitivity analysis is needed. We also agree with reviewers #2 and #3 that a comparison to temperature-index models in addition to the comparison with the linear machine learning model is necessarily to reinforce the broad applicability of the proposed method. Please also take care to address the physical processes underpinning the noted non-linearities (Reviewer #3). While the regionality of the study's scope does not preclude publication, we would also like to see more discussion, and if feasible analysis, of how the proposed model would apply to other glacial regions, echoing Reviewer #3's comments.

If you feel that you are able to comprehensively address the reviewers' concerns, please provide a point-by-point response to these comments along with your revision. Please show all changes in the manuscript text file with track changes or colour highlighting. If you are unable to address specific reviewer requests or find any points invalid, please explain why in the point-by-point response.

We would like to thank the editor and reviewers for their insightful and particularly constructive comments. Following their suggestions, we have performed a vast revision of the manuscript, updating the main story line, running new simulations, adding new plots and completely rewriting the Discussion and important parts of the Results sections. We believe these changes add more depth to the findings of the study, giving more insights on the potential physical processes involved in the nonlinear climate-glacier interactions, and expose why these findings matter for other glacierized regions. We have performed a whole new set of experiments in order to show that these nonlinear mass balance effects are very relevant for ice caps, further developing the initial arguments and providing a more complete picture of the role of nonlinearities for global glacier models. Moreover, following a lengthy literature review, we argue that the nonlinear response to future warming by our nonlinear model can be explained by the lesser role of shortwave radiation in the glacier surface energy budget in warmer future climate scenarios. Models with linear relationships between PDDs and melt are known to be over-sensitive to temperature

changes. We demonstrate that both the linear Lasso MB model and the temperature-index model from GloGEMflow follow this behaviour, and we point out the need to improve MB models for ice and snow projections under climate change, particularly for flatter glaciers and ice caps. New parameterizations are needed to accommodate the reduction in degree-day factors (i.e. melt sensitivity) to air temperature expected in the future. We believe these new findings are relevant to a wide audience, ranging from glacier and ice sheet modellers to snow and mountain hydrologists. Following these new interesting findings, we have decided to slightly shift the focus of the paper towards the nonlinear climate-glacier interactions instead of the French Alps results. This has resulted in some changes in the title and the abstract, while keeping the first sections of the Results on the French Alps unchanged.

Reviewers

Reviewer #1 (Remarks to the Author):

Dear Authors

The manuscript titled "Deep learning unveils non-linear climate-glacier interactions through the 21st-century deglaciation of the French Alps" is exciting. The manuscript is an excellent contribution towards Glaciology/climatology.

I have some comments to improve the quality of the manuscript.

1. Page 12, Line 206: Authors refer to models from previous publications; however, it will be significant if authors can provide information on model architecture, training information, parameter optimization, etc.

We have added additional details on the architecture and hyperparameters of the chosen neural network in the methods section following the comments by the reviewer. We now cover the neural network architecture, optimization algorithm, parameter initialization, regularization techniques, activation functions and batch normalization.

2. Have authors tried Shallow learning or machine learning models? Authors should add information about recent shallow neural networks/ML/DL applied in glaciology.

Depends on what we understand by Shallow learning. If you refer to feeding the model with predefined features instead of letting the model find them by itself, in our case we performed a sensitivity analysis to choose the most relevant climate and topographical predictors for glacier mass balance. Alternatively, if you refer to the depth of the model architecture, yes, shallow neural networks have been tested. The chosen (deep) architecture is the smallest possible architecture that delivers the best possible performance for the training dataset in cross-validation. Additionally, in Bolibar et al. (2020) a thorough comparison with the Lasso was performed, showing

major gains when switching to a more complex nonlinear statistical model like a deep neural network.

A new paragraph has been added in the introduction providing additional context in the state of the art of neural networks for regression problems in glaciology. A more thorough review of previous papers applying machine learning to glaciology can also be found in Bolibar et al. (2020).

3. The word Deep Learning seems to be overused; instead, the model name should be given because Deep Learning itself is an extensive area comprising LSTM, CNN, hybrid models, etc.

Technically, the model is a Deep Feed-forward Multilayer Perceptron, but this is not the way it is often mentioned in the literature, so we preferred to use the shorter term Deep Learning. We have specified the exact model type in the Methods section, and we have kept the shorter term Deep learning in the main text in order to be more concise.

4. The methodology section is not providing the core of models, which is a weaker part of this manuscript.

We have added a new paragraph with more details on the neural network architecture (as required by remark #1). It provides an overview of the model design, but we have a full publication dedicated to explaining the model details (Bolibar et al., 2020). For the more avid reader interested in these details, we believe it is better to point towards that paper in order to avoid duplicating too much information between these two publications.

5. Page 14, Line 393: What is how-water drilling?

That is indeed a typo. It has been corrected to "hot-water drilling".

6. Page 16 Line 467-472: "A linear regression model....ice volume in 2015." How can a linear model provide reliable estimates for such a non-linear behavior of glacier ice volume forecasting? Have the authors tried the sensitivity analysis of this regression?

Linear models can approximate nonlinear behaviours, but they will just perform worse than a suitable nonlinear model. A sensitivity analysis of this linear regression is presented in detail in Bolibar et al. (2020), and is summarized in Fig. 5 of that paper. Additionally, we also performed an error analysis of that model, which is available in the supplementary materials of that paper, and which is summarized in Fig. S1 of the same.

The manuscript requires some brief yet essential information on the model characteristics, as discussed in earlier comments.

Through our previous replies to the reviewer's comments we have addressed this aspect. We have extended the explanations on the model details in the Methods section, in order to provide the necessary background to the reader. For all the rest of the details we have a fully dedicated publication (Bolibar et al., 2020) dedicated to the methods.

Reviewer #2 (Remarks to the Author):

General comments

This manuscript uses a previously-established deep learning approach of simulating glacier change (mass balance and ice dynamics). I am not an expert in deep learning or machine learning, but I see that the method is already published in Bolibar et al., 2020, so hopefully has already been assessed by experts. Here, the deep learning method is applied to simulating glacier change in the French Alps through the 21st century. These future projections using the deep learning method are then compared to results from a linear machine learning model to better understand non-linear components of the glacier-climate system. This component of the paper- investigating the impact of linear versus non-linear models on glacier mass balance, is interesting and an important contribution to glaciology. As the authors note, the majority of future glacier projections currently use temperature index models, so better-understanding weaknesses or biases in these models is important. The paper is well written, especially the introduction, and nicely ties in the big picture/larger importance of this work in the main text and especially in the Supplementary Information text.

In the paper, linear mass balance models are associated with temperature-index models (e.g. ling 28). However, the linear mass balance model that is compared with the deep learning approach is another machine learning model (LASSO). A few suggestions around this point include: 1) It should be clear that the results are being compared with a machine learning approach and not a traditional temperature-index model. 2) I would have liked to see the results of the deep learning approach also compared with the temperature-index model, which most glaciologists use and or would understand. Right now, results are compared with results of GloGEMflow, but with the only analysis being Fig S5. 3) I also wish there was a brief description of the LASSO model in this paper when it is first introduced, right now the authors reference a previous paper.

We are grateful for this feedback. These different comments have helped us to further investigate the analysis presented in this paper, reaching more mature and meaningful conclusions. We respond to the different remarks by the reviewer individually, covering the changes made to address them:

(1): In the results section we have performed separate analyses for the Lasso mass balance model and GloGEMflow's temperature-index model. These are placed in separate paragraphs, with the main analysis being performed with the linear machine learning model. The main reason for this is because ALPGM allows switching mass

balance models, thus perfectly isolating the effects of mass balance nonlinearities. After presenting these results, we present the same analysis between ALPGM's deep learning model and GloGEMflow's temperature-index model. Despite not having the figures presented in the main text, the simulations and analysis are exactly the same as the ones between the Lasso and the deep learning model (Fig. S5 vs. S6). The main goal of that section in the main text is to show that even for a more complex temperature-index model with several melt factors (e.g. for snow, firn and ice) as the one used in GloGEMflow (in comparison to some global model using one melt factor and monthly temperature), the behaviours are similar to those from a linear statistical model. Therefore, the goal of that section is to highlight the similarities between our findings with machine learning models and temperature-index models. From that point, in the discussion, we sometimes refer to these models as "linear models", in order to differentiate them from the nonlinear deep learning model. However, we often refer to them as "models with linear relationships between PDDs and melt, and precipitation and accumulation". Indeed, one might argue that a temperature-index model with two melt factors is not exactly linear. In order to avoid this sort of ambiguity, we have now added additional explanations regarding this aspect throughout the text. We argue that the behaviours of these temperature-index models with two melt factors can also be extrapolated to what we observe with our linear statistical model, despite not being exactly linear. We demonstrate that in fact our linear Lasso MB model is equivalent to an even more complex temperature-index model, as it has the equivalent to 13 degree-day factors instead of the three found in GloGEMflow's temperature-index model. As we argue in the new Discussion section, the linear vs. nonlinear MB responses do not arise from the number of degree-day factors, but from their linearity (being a simple scalar parameter for the Lasso and temperature-index) and their nonlinearity (the response of melt and glacier-wide MB to changes in air temperature and snowfall varies nonlinearly as air temperature and snowfall increase or decrease in the future).

(2): As mentioned in the previous paragraph, the analyses between the deep learning model and GloGEMflow's temperature-index model are made in exactly the same way as the one between the two machine learning models (Fig. S5 vs S6). In the Results section from the main text, we focus on showing how similar the behaviours from both models are through figures S5 and S6. Moreover, Fig. S10 (previously S8) also compares the mean glacier altitude between both models, and Table S2 gives many details on the different characteristics of ALPGM and GloGEMflow. In order to extend this analysis, we have now added a new figure (Fig. S7) comparing the glacier ice volume evolution. The results follow the same conclusions drawn from comparing Fig. S5 and S6, and give additional support to the Results section.

(3): We have added an explanation of the Lasso in the methods section, following the extended explanations of deep learning model requested by Reviewer #1.

A main finding of the paper is that deep learning captures important nonlinearities in the glacier/climate system. The authors nicely investigate this concept and present the results. It is interesting that the impact of the different approaches on cumulative mass balance is very minor (Fig 4j). I think it needs to be clear that while the deep learning approach does improve the simulation of extreme positive and negative mass balances, the results here

don't show a huge difference on cumulative mass balance. A discussion point could be added that centers around the most appropriate problems to apply a deep learning approach (when capturing extreme extreme glacier mass balance years is important) versus when a traditional temperature-index model is sufficient (when capturing cumulative glacier mass balance is the goal, at least suggested by these results).

This is one of the aspects that has been completely rewritten in the manuscript, also to consider some comments of Reviewer #3. We have further investigated the role of the mass balance nonlinearities by performing additional synthetic experiments to approximate the behaviour of ice cap type glaciers (e.g. no topographical adjustment). For that, we keep the glacier geometries constant (and so their centroid and therefore the altitude where the climate data are quantified) to compute their mass balance. This virtually eliminates the positive topographical feedback from mountain glaciers retreating to higher elevations, thus producing more extreme negative mass balance rates throughout the 21st century. We showcase that for such glaciers the mass balance nonlinearities play a much more important role. This is all the more interesting as ice caps happen to store a large percentage of all ice present outside the two main ice sheets (Greenland and Antarctica) and that their future evolution is of paramount interest for sea level rise. Alternatively, we also argue that for glaciers with a strong topographical feedback (lying on steep slopes), and therefore with a rather short response time, linear mass balance models (e.g. temperature-index) are adequate tools. Such glaciers can adjust their geometry to the current climate, thus avoiding the extreme warming by the end of the century. Fig. 4 has been redesigned to explain this, and a new figure (Fig. 5) has been included to better explain the relationship between glacier geometry and nonlinear climate-glacier interactions. Moreover, following the comments by Reviewer #3, we have performed a thorough investigation on the potential physical processes behind the captured nonlinear MB responses. Therefore, we argue that these findings are particularly important for flatter glaciers and ice caps, and that they should be addressed with nonlinear degree-day factors (via new parameterizations) or by using more complex surface energy balance models (which are currently too complex and need large amounts of meteorological data at high spatial resolution as input to be used at large geographical scales).

I would have liked more discussion on the description and significance of Figure 4. This is the main Figure showing the importance of the deep learning approach to capturing nonlinearities, which I think is the biggest contribution of this paper.

As we have explained in the previous paragraph, this aspect has been completely rewritten and now there is a much more in depth analysis in the Discussion section. Fig. 4 has also been redesigned and a new figure 5 has been added.

Specific comments

L62 - "strongest glacier retreat" not clear what strongest means, fastest?

We have updated the sentence to "... fastest glacier retreat ..." as suggested.

Fig 1 - Why not have Figures S1 & S2 here? The supplementary figures are useful in showing all 3 RCP scenarios, which would be nice to include in the main text.

The main reason we chose to keep Figures S1 and S2 in the supplementary materials is the fact that they are based on only nine climate scenarios. Fig. 1 is instead based on 29 climate scenarios, providing a much more robust and significant signal. Moreover, the main message of the paper regarding the future evolution of French Alpine glaciers can perfectly be conveyed without having to show those nine climate scenarios, thus keeping the message and the main text leaner and easier to read.

L159 - not clear what the correlations are for, modeled vs measured glaciological mass balance? Modeled vs measured PDDs and snowfall (from what climate data)?

Indeed, it was not too clear what the correlations were based on. They are actually based on cross-validation, with respect to observations used to train the models. The fact that this comes from cross-validation is now specified in the text, and a citation has also been added for context.

Fig 3 - Would also be interesting to see climate anomalies as a %. For example, -1200 or +1200 CPDD anomalies seem like huge differences from normal. The extreme values are where the two models diverge, is there a way to show how often anomalies fall in those extreme values.

This is a good point. It is hard to establish a reference value to compute the anomalies as %, as there are many different glaciers and climate scenarios involved. In order to represent where anomalies fall in those extreme values, we have computed the mean value and the standard deviation of all maximum and minimum climate extremes for the three represented mass balance forcings in Fig. 3. These have been computed for all 29 projected climate scenarios, and give additional context on where the majority of the climate extremes will occur through the 21st century in the French Alps. We have added this information in Fig. 3, which helps to better constrain the influence of each forcing in the 21st century MB projections.

Figure 4c,f,i - does the deep learning model never result in positive mass balance? Figures a,d,g show there are years with positive mass balance. Why are the PDFs all negative?

The PDF is based on the averaged signal of all glaciers in the French Alps and from all climate scenarios for a given RCP (13 for RCP 4.5 and 8.5, 3 for RCP 2.6). The deep learning mass balance model does simulate positive MB rates, but when the simulations of multiple glaciers and climate scenarios are averaged, the extreme (positive and negative values) are "smoothed", giving a signal representative of the average conditions. Since the MB signal depends on glacier location/topography and on climate scenarios, performing this sort of averaging with such large amounts of data helps capturing a clearer and less noisy signal from these comparisons. The spread is so large among climate scenarios that looking at the signal(s) without

averaging makes it very hard to discern trends. We have updated the legend of Fig. 4 in order to clarify how the PDFs are computed.

L297 - 'remarkable' is a bit opinionated, is there a better word?

This has been updated to "relevant".

Lauren Vargo

Research Fellow

Antarctic Research Centre

Victoria University of Wellington

Reviewer #3 (Remarks to the Author):

This paper applies a data science method of modeling glacier mass adjustment to climate in the French Alps for the coming century when they drive the model with an ensemble of multiple future models under three emission scenarios (RCP 2.6, 4.5, 8.5). The extent and nature of the predicted glacier mass balance (MB) response is alarming: by end of this century, fully 75 to 88% of the glacier volume will be lost, and what remains will be dramatically altered landscapes. It is really intriguing to see the disappearance of mean surface area, emphasizing loss of the smaller and lower glaciers. It is also eye-opening to see that cumulative MB differences were mitigated by a complete loss of half the glaciers by mid-century. The unique contribution the authors claim is in the methodology: that this deep learning approach improves upon linear mass balance modeling.

The authors champion their neural network (deep learning) method as being an important new contribution for the scientific community, for which they provide a more complete cross-validation in a prior paper using the same dataset. They demonstrate that deep learning does give "superior nonlinear explained variance," making models more accurate because both climate and glacier systems are known to react non-linearly to forcing. This is especially important as glaciers and climate deviate more extremely from mean conditions that appear most often in calibration. In this paper, they focus on the results of running model with 29 different climate members. They also provide a test of this relative mass balance response to linear vs nonlinear idea by comparing linear regression model to non-linear deep learning (artificial neural network) model of MB. They use the same predictors that are both topographic and climatological. They show that for more extreme values, the non-linear model makes a profound difference. However, the problem here is that for most of the 21st century, there is very little difference in the models; linear and nonlinear MB simulations are nearly identical.

In assessing the merit and impact of this work, one wonders about the appropriateness of this national focus on just a small subset of glaciers in France only. Are these new insights of

sufficient scope? What new insights do we gain about the future of Alps? And what about other regions where the authors suggest such a modeling approach would be more important? Does this method improve what would be predicted by a more temperature dependent index method that the authors categorize under the generalized 'linear' methods? Furthermore, what are the actual physical process insights elucidated by this approach? As a data-driven method, it is not solving fundamental equations of dynamics.

These are all very interesting questions, which have driven the main changes in the refactoring of the manuscript for this new version. We believe adding these aspects into the manuscript has helped improve it, adding more depth and providing a more solid main story line regarding the implications of nonlinear climate-glacier interactions for global glacier models. These changes have particularly helped to identify the limitations of models using linear relationships between PDDs and melt, and precipitation and accumulation for flatter glaciers and ice caps. And these comments also lead us to a lengthy literature review, enabling us to determine that the captured nonlinearities arise from a decreasing sensitivity of MB to increasing temperatures under climate change. We will cover each of the changes performed in the manuscript following each of the reviewer's comments.

Regarding the geographical focus on the French Alps, as we previously discussed, it is not straightforward to determine if the behaviours observed in the French Alps will be relevant in other glacierized regions. In order to further investigate this subject, we have performed a whole new set of synthetic experiments in which we explore the role of mass balance nonlinearities for what we have called ice-cap type glaciers, i.e. glaciers for which a topographical feedback is not possible, as it is the case of mountain glaciers. It is noteworthy that ice caps store a large percentage of all ice outside the two main ice sheets. Regions like Arctic Canada and Russia include a large amount of ice caps, for which uncertainties in projections are very high. To do so, in this synthetic experiment, we performed simulations keeping the glacier geometries constant throughout time, thus keeping the glacier centroid and therefore the altitude at which we obtain the climate data to compute the mass balance constant throughout time. This virtually removes the positive topographical feedback of steep mountain glaciers, and allows investigating the response of ice caps to future climate warming. We show that for such glaciers, mass balance nonlinearities play a much more important role. Ice caps cannot retreat to higher elevations, thus enduring much higher air temperatures and consequently more extreme negative mass balance rates throughout the century. In order to further explain this, we have completely rewritten the Discussion section and large parts of the Results. Moreover, Fig. 4 has been remade by comparing the behaviour of mountain glaciers vs. ice-cap type glaciers. A new figure (Fig. 5) has been added, which summarizes the main findings of the paper regarding glacier geometry, topographical feedback and nonlinear climate-glacier interactions.

Regarding the future of the Alps, this study improves projections of glacier evolution as it has used all currently available data in the French Alps, which was not used in any previous study covering the region (e.g. Zekollari et al., 2019). Additionally, the skill (both RMSE and r^2) of the mass balance model is better than any other study, which combined with the rich training dataset mean that these projections carry

reduced uncertainties compared to previous studies. Region-wide simulations in terms of volume and area do not differ much (except for RCP 2.6, for which our estimates are significantly more pessimistic). Nonetheless, simulations for individual glaciers which are relevant for touristic, hydrological and cultural reasons are much more accurate, giving important information to French society and decision-makers on the magnitude of the impacts of climate change to iconic French glaciers.

Regarding the potential physical processes explaining this, it is true that such a "black box" neural network offers limited explanations on the origins of these simulated differences. Despite the "black-boxness" of neural networks, we managed to extract the climatic response of the model to the main drivers of glacier mass balance. In order to understand and attribute the captured nonlinear sensitivities of our MB models to different climate forcings, we performed an extensive literature review on the physical basis and limitations of temperature index models. Already in 1995, Roger Braithwaite (1995) pointed out that temperature-index models provide a very simplistic response to changes in air temperature. He demonstrated with both observations and more complex surface energy balance models, that the response of degree-day factors associated to different albedos and turbulent fluxes vary significantly with air temperature. This is particularly important for low albedo values typical of ice, implying that the sensitivity of ice melt to increasing temperature decreases strongly, which is something that cannot be captured by linear models like the Lasso or temperature-index models. Another study by Pellicciotti et al. (2005) confirmed the same results with an alpine glacier in the Italian Alps, highlighting the "over-sensitivity of temperature-index to changes in air temperature". Finally, a study by Huss et al. (2009) further confirmed this, for which they detected a decrease of -7% per decade in degree-day factors for snow during the ablation season. They did not perform the same analysis for ice degree-day factors, but after our analyses we can only expect the differences to be even larger. That study concluded saying: "We find relatively stable DDFs until the mid-1970s followed by a negative trend of 7% per decade (Figure 4d). The drivers for these long-term variations cannot be detected based on the available data sets as they do not resolve all components of the energy balance. Higher air temperature dependent incoming longwave radiation (Figure 4c) plausibly explains part of the observed decrease in DDFsnow over the last decades, confirming an oversensitivity of temperature-index models to temperature change [Pellicciotti et al., 2005]. DDFs are, however, also affected by variations in global shortwave radiation, and, to a lesser extent, by turbulent heat fluxes [Braithwaite, 1995; Ohmura, 2001]. We therefore caution against using classical temperature-index models calibrated in the past for projecting snow and ice melt in glaciological and hydrological studies and to calculate future sea level rise." This coincides with the findings by Braithwaite (1995), with a strong nonlinear decrease of ice melt sensitivity to increasing air temperatures. Furthermore, these results correspond perfectly with the captured nonlinear response of our deep learning MB model. For the most extreme climate scenarios (RCP 2.6 for mountain glaciers, all climate scenarios for ice caps), the nonlinear model is clearly less sensitive to changes in air temperature. Moreover, this reduced sensitivity from the ablation season is captured in Fig. 3c, with the reaction to summer snowfall depicting a reduced sensitivity compared to the Lasso. This analysis also accommodates other particular behaviours observed in our machine learning models. In cross-validation,

the Lasso was found to be more biased for extreme MB rates, giving a contradictory behaviour with what is observed in the projections (Fig. S9 vs. Fig. 3 and 4). This behaviour can also be explained by the linear sensitivity to changes in air temperature of the Lasso. The Lasso is calibrated with observed air temperature changes between 1967-2015, already giving worse results than the nonlinear model for this past calibration period in cross-validation. However, its over-sensitivity to temperature changes implies that for future projections the bias observed in cross-validation is reversed: the bias goes from providing estimates that are under-sensitive to extremes for past periods (Fig. S9), to providing estimates that are either too high for the positive MB rates or too low for the negative MB rates for future projections (Fig. 4).

The lack of insight on the underlying physical processes when working with neural networks is a well known problem by the machine learning community (e.g. Rackauckas et al., 2020). In order to go a step further the limitations of this study, we have started a new project combining differential equations of glacier processes (e.g. enhanced nonlinear temperature-index model and the Shallow Ice Approximation) and neural networks. Such an approach aims at precisely identifying the governing processes driving large-scale glacier changes. Therefore, this study clearly does not provide all (but some) answers to what these nonlinearities are exactly due to, but it serves to raise awareness of their importance, it quantifies them, and points out new research avenues to be explored with more complex models like the one just mentioned. This aspect has been added in the Discussion section, in order to give context on the relevance of these findings and what are the next steps to be investigated.

I'm persuaded that this is a novel method applied to glacier change studies, but I lack adequate knowledge to give thorough technical review of the deep learning. I think that a more compelling case could be made to justify this data-analytical method by showing how it better captures physical processes, or allows for specific new predictive capacity for water management in the future. I think more discussion about the specific 'non-linearities' and their implications would help. The non-linear deep learning is not obvious to the non-expert. As written, the authors show how mass balance responds non-linearly to some specific variables in Fig 3, but don't explain why in terms of physical processes. It is insufficient to claim that climate and glacier systems are known to be nonlinear (L149). What does it mean to treat PDD in a non-linear way? Are there feedbacks or sensitivities of mass balance to specific variables that are characterized by nonlinear models? If so, how? Furthermore, are they really that big of a deal when it seems that we won't see much difference for 50 more years, and by then there will be so much less ice anyway that it is perhaps "much ado about nothing"? As written, the paper suggests that uncertainties "might be even greater than we previously thought" for future climate scenarios, but does not hypothesize why or where. The authors claim that their work is very similar to the linear models, and only diverge in extremes. They speculate that the impacts could be more pronounced in complex regions. However, this is hard to grasp, since the method is effectively as good as the data.

This study raises awareness on the importance of nonlinearities in glacier mass balance. Since we use statistical models, it is not possible to directly attribute changes to specific physical processes, but it serves to quantify them and to

understand the dynamics. Fig. 3 clearly shows the nonlinear response to climatic drivers, and as we argued in the previous paragraph, these differences can potentially be explained by known nonlinear physical processes related to degree-day factors and variations in air temperature. We have added a new section in the Discussion in which we explain the physical processes that can explain these nonlinear mass balance responses. Additionally, we demonstrated that a nonlinear MB model is more skilled than a linear model in representing annual mass balance rates. This means that glacier evolution projections, and therefore hydrological studies from glacierized basins, will have reduced uncertainties compared to projections done with linear mass balance models.

We agree with the reviewer that the previous version of the manuscript did not give enough details on why and where these nonlinear mass balance responses could play a more important role. In order to further explore this aspect, we performed new simulations reproducing the behaviour of ice caps, and we showed the enhanced importance of these nonlinear sensitivities for such cases. There is a very interesting relationship between glacier geometry and mass balance nonlinear responses. Ice caps have a rather negative topographical feedback, since when their thickness decreases they move to lower altitudes, further enhancing the melt. This produces more extreme negative mass balance rates, which are associated with an increased importance of nonlinear MB effects. We have added a whole new section in the Discussion about this, as well as a new version of Fig. 4 and a new Fig. 5. This new experiment identifies two main behaviours in mass balance nonlinearities: (1) Mountain glaciers lying on relatively steep terrain and with short response times can adequately be simulated with a linear mass balance model. (2) Ice caps and large valley glaciers with long response times are better simulated with nonlinear mass balance models. This gives new insights to the global glacier modelling community, pointing out the regions which have increased uncertainties and that will require more complex models, either statistical nonlinear models or physical models including some of the nonlinear processes discussed in the manuscript.

It is interesting to see how initial ice thickness is the main source of uncertainties of glacier simulations. Only 4 of the glaciers have such data. Is there a need to get more of these measurements?

This is one of the main challenges of large-scale glacier modelling and the topic of a lot of current research. There is indeed the need for more observations of glacier ice thickness, but such observations are very costly to obtain. They require large teams of people to perform radar measurements on complex terrain. This means that only very developed nations like Switzerland can afford to have an important percentage of their glaciers measured. The GlaThida database gathers all observations performed worldwide. In this study, we used all the available observations in the French Alps, performed by the GLACIOCLIM national glacier observatory.

Some line edits:

L23: The abstract mentions 75 and 88% glacier volume loss by end of century. Loss from what point, the 2021 currently existing volume? Explain/clarify.

The glacier volume loss is with respect to the year 2015. Since there is a very strict limit in the number of words in the abstract, we prefer to omit that in order to keep other more relevant findings in the abstract. The year is clearly mentioned throughout the manuscript.

L50: delete the 'in order' before to

The sentence has been updated as suggested.

L53: type of model

Updated as suggested.

L103-106: The description of the future snowfall rates on glaciers not being effected is not clear as written. Maybe articulating more clearly about elevation dependent ratio of solid to liquid precipitation would make this more evident.

A new sentence has been added in order to clarify this aspect: "The increase in glacier altitude also helps the solid to liquid precipitation ratio to remain relatively constant."

L137: would be better worded as, "...Little Ice Age that are strongly out of balance with the current climate."

The sentence has been updated accordingly.

L138: change wording; our projections describe the almost COMPLETE disappearance

Updated as suggested.

L140: instead of store, I recommend "retain"

Updated as suggested.

L141: large should be "larger" and give a dimension to match the Fig S4 (>2 km²)

Sentence updated accordingly.

L187: rely should be "relies"

Updated as suggested.

L228 : change are to "comprise"

This sentence has been removed from the new version of the manuscript.

L242: typo? Not sure what is Fig. "4Sb"

Indeed, it was a typo. This sentence no longer appears in the new version of the manuscript.

L246: also, 5Se

Same, a typo. This sentence has also been removed from the new version.

L268-270: this seems a stretch beyond the data to speculate that heterogeneous topography and "climatic char's" are likely to go thru higher variety of climate extremes. On what basis?

This sentence has also been removed from the new version of the discussion. See previous comments for our justification on why this might be relevant for other glacierized regions.

L274: sentence ends incompletely.

This sentence has also been removed in the new version.

L275-278: awkward word choice/prose: "in a first term" and "in a second term." Do the authors intend this as an idiom like, "on one hand," or is it referring to an actual quantitative measure of uncertainty term? Please clarify.

This is indeed used as a quantitative measure, with the "first term" implying the first and most important source of uncertainty. The sentence has been rephrased as follows in order to make it smoother:

"The main uncertainties in future glacier estimates proceed from future climate projections and levels of greenhouse gas emissions (differences between RCPs, GCMs and RCMs), whose relative importance progressively increases throughout the 21st century. With a secondary role, glacier model uncertainty decreases over time, but it represents the greatest source of uncertainty until the middle of the century."

L354: should be combined effects

Updated as suggested.

Fig. 2: (a) confusing shading below the 2015 mean elevation; makes it seem like topography rather than median elevation.

This is the intended effect. The glacier mean elevation is very related to the mean elevation of each massif. The vertical orientation helps understand the topography of the different massifs and how glaciers will evolve in them throughout the century.

In c - e panels, lines all have end points; is that when respective glaciers disappear? Should be mentioned/clarified.

The following sentence has been added in the legend: "Years in white in c-e indicate the disappearance of all glaciers in that massif."

Also, there appear to be random pixelated yrs of positive MB in (c). Why?

As glaciers retreat upslope, it is not impossible for them to have slightly negative to slightly positive MB (as indicated by the legend). This is perfectly normal, particularly in the northern massifs with lower temperatures and higher precipitation rates.

Caption to Fig. 2 (c) has a spurious "A"

This has been removed as suggested.

Fig. 4: the RCP 8.5 graphs are strange. Annual nonlinear difs starts in positive values. How?

Even if both MB models have been calibrated with exactly the same data it does not mean they will produce exactly the same results. The models have been calibrated with past climate data, and the dataset used for the climate projections is based on random statistical trajectories set by different levels of greenhouse gases emissions. Moreover, every year quite a lot of glaciers disappear, thus changing the simulated state from the past states used to train the MB model. As glaciers disappear, certain characteristics of each MB model (nonlinearities due to extreme forcings) will show up differently, enhancing the differences. The very first year produces almost exact results, and after that year the results slightly drift away.

And the cumulative nonlinear difference goes positive. Is that right? Explain.

The slight differences explained above produce slightly more positive results for the deep learning model, thus giving a positive difference in cumulative.

Table S1: climate 'members' to force glacier evolution model

What should exactly be changed? This seems like the current legend of the table.

Reviewers' Comments:

Reviewer #1:

Remarks to the Author:

Dear Authors

I am happy and satisfied with your efficient efforts to address all the concerns. I hope this manuscript will be a milestone in glaciological studies reaping the benefits of AI/ML/DL.

Thanks

Anul

Mohd Anul Haq

Reviewer #2:

Remarks to the Author:

The updated version of this manuscript addressed all of my original comments. I enjoyed reading the updated version, and I think that this paper is a valuable contribution to science. The new, additional analysis and discussion around feedbacks, glacier geometry and topography, and the impacts of these results strengthens the manuscript. My only note is that some paragraphs are quite long. I wonder if some can be split up, making them easier to read.

Reviewer #3:

Remarks to the Author:

The authors have undertaken a significant revision to the original manuscript that addresses many of the concerns raised by myself and other reviewers. Congratulations. I think the additional figures, amplified discussion, and more complete literature review all broaden the relevance of this study, justifying the title change and making it more generalizable. Yet there are now implications for modeling ice caps that emerge in discussion that I want to be sure do not go beyond what the model can support. The non-linear deep learning model here is developed and applied to the French Alps. The authors articulate well how their model does show decreased sensitivity to extreme conditions, and then show a compelling difference between mountain slope glaciers and flat ice caps. This provides the opportunity to make more generalizable claims about different responses of mountain glaciers and ice caps. However, I am unsure how appropriate the claims are that suggest ice cap models under higher RCP scenarios are over-estimating mass loss and sea level rise. Since it is a statistical model tuned to mountain data, is it too speculative to comment on polar ice caps and sea level rise, since the polar ice cap stability is arguably much more tied to other dynamics (i.e. ocean energy, temperature)? If the authors can explain this briefly, it would make the paper stronger.

I found it somewhat confusing to follow how the discussion treats the biases revealed by the model comparisons. On L305-08 the authors suggest that non-linear MB changes would be more negative, even while Fig. 4e shows the opposite -- the linear models showing more negative MB. Is the text misleading here? e.g. "flatter glaciers and ice caps will experience substantially more negative MB rates...and therefore greater differences due to nonlinearities for the vast majority of future climate scenarios." I may be misunderstanding the text, but I just want to be sure.

This seems to be rectified later in the discussion of consequences of these biases. L336 mentions that indeed the linear models have a "tendency to negative MB biases." However, would it be worth discussing why? What processes would explain this? This implies more optimistic scenarios of ice cap preservation. Nevertheless, a model predicting less responsive ice cap mass balance decrease is actually counterintuitive. The schematic in Fig. 5 depicts that without vertical displacement, the ice

mass is vulnerable to increasingly negative mass balance. So what processes could explain the lower sensitivity seen in the model? Some clarity here would make the paper stronger.

I like the new Fig. 5. However, two other issues: First, I find myself mixed up about positive and negative feedbacks. Wouldn't the retreat to higher altitudes of the mountain glacier case be depicting a negative feedback (i.e. counter acting the nature of the initial perturbation, while the thinning ice cap is a positive (i.e. self-reinforcing), feedback? Second, the figure states "nonlinear mass balance sensitivity" following "+extreme negative mass balance." Yet shouldn't the figure more clearly show that the model predicts less mass loss than the linear model?

Minor stylistic points: It might make the discussion easier to read by breaking up that first immense paragraph (i.e. end of L259).

Is "parallelisms" a good word choice? I find it arcane, and reminds me of grade school grammar class.

Overall, a very interesting and well written paper. Hopefully my ambiguities are merely my lack of understanding of the method.

Bolibar et al. responses to editor and reviewer's comments

Author responses are given *in italic* after each of the editor and reviewer's comments.

REVIEWERS' COMMENTS

Reviewer #1 (Remarks to the Author):

Dear Authors

I am happy and satisfied with your efficient efforts to address all the concerns. I hope this manuscript will be a milestone in glaciological studies reaping the benefits of AI/ML/DL.

Thanks

Anul

Mohd Anul Haq

Thanks a lot for the positive feedback, we appreciate the comments given throughout the review process.

Reviewer #2 (Remarks to the Author):

The updated version of this manuscript addressed all of my original comments. I enjoyed reading the updated version, and I think that this paper is a valuable contribution to science. The new, additional analysis and discussion around feedbacks, glacier geometry and topography, and the impacts of these results strengthens the manuscript. My only note is that some paragraphs are quite long. I wonder if some can be split up, making them easier to read.

We are grateful for the positive feedback to our modified version of the manuscript. We believe the constructive comments given in the first round greatly helped to improve it and push it towards its current version. , Following the reviewer's suggestion (which was also made by Reviewer #3), we have split several paragraphs in order to improve readability.

Reviewer #3 (Remarks to the Author):

The authors have undertaken a significant revision to the original manuscript that addresses many of the concerns raised by myself and other reviewers. Congratulations. I think the additional figures, amplified discussion, and more complete literature review all broaden the relevance of this study, justifying the title change and making it more generalizable. Yet there are now implications for modeling ice caps that emerge in discussion that I want to be sure do not go beyond what the model can support. The non-linear deep learning model here is developed and applied to the French Alps. The authors articulate well how their model does show decreased sensitivity to extreme conditions, and then show a compelling difference between mountain slope glaciers and flat ice caps. This provides the opportunity to make more generalizable claims about different responses of mountain glaciers and ice caps. However, I am unsure how appropriate the claims are that suggest ice cap models under higher RCP scenarios are over-estimating mass loss and sea level rise. Since it is a statistical model tuned to mountain data, is it too speculative to comment on polar ice caps and sea level rise, since the polar ice cap stability is arguably much more tied to other dynamics (i.e. ocean energy, temperature)? If the authors can explain this briefly, it would make the paper stronger.

We are grateful for the positive feedback to our updated version of the manuscript. We believe it was greatly helped by the constructive comments given during the first round of reviews.

We understand the concerns on drawing conclusions on ice caps based on a synthetic dataset from mountain glacier data. Indeed, one should be careful when taking into account the conclusions from our synthetic experiment, since they are focusing on just one of the processes of glacier change. For marine-terminating ice caps, calving and ocean temperature will also play a role, as stated by the reviewer. Nonetheless, mass balance-related changes are expected to play an important role as well for these regions (see figure below). We added the following sentence to further clarify this: “These conclusions drawn from these synthetic experiments could have large implications given the important sea-level contribution from ice cap-like ice bodies (Marzeion et al., 2020). However, to further investigate these findings, experiments designed more towards ice caps, and including crucial mechanisms such as ice-ocean interactions and thermodynamics, should be used for this purpose.”

Figure 3. Specific mass balance rate in $\text{kg}\cdot\text{m}^{-2}\cdot\text{yr}^{-1}$ as a function of time and RCP scenario. Shading indicates ± 1 standard deviation of ensemble (shown only for RCP2.6 and RCP8.5 for clarity). Solid lines indicate ensemble median. Dotted lines in global panel indicate averages of all regional ensemble medians, weighted by glacier area of the respective region. Regional panels are sorted by descending regional glacier mass here and in all following plots. Size of ensemble and number of glacier models (in brackets) are shown in the lower left corner. Figures S1–S4 show the results for the individual glacier models.

I found it somewhat confusing to follow how the discussion treats the biases revealed by the model comparisons. On L305-08 the authors suggest that non-linear MB changes would be more negative, even while Fig. 4e shows the opposite -- the linear models showing more negative MB. Is the text misleading here? e.g. "flatter glaciers and ice caps will experience

substantially more negative MB rates...and therefore greater differences due to nonlinearities for the vast majority of future climate scenarios." I may be misunderstanding the text, but I just want to be sure.

In the above mentioned sentence we are not comparing the linear and nonlinear mass balance models. After presenting the synthetic experiment, which consists of modified simulations of alpine glaciers, we present how these new synthetic simulations of "ice caps" compare to the mountain glacier ones. The goal of that sentence is to describe the consequences of keeping glacier geometry constant through time, which is quite similar to what many ice caps will go through (particularly in terms of surface area). There are two main topics analyzed in this paper: (1) the main one is the role of mass balance nonlinearities due to climate forcings, and (2) a secondary topic is the role of glacier geometry, which relates to the previous one.

In order to make this clearer to the reader, this sentence has been rephrased.

This seems to be rectified later in the discussion of consequences of these biases. L336 mentions that indeed the linear models have a "tendency to negative MB biases." However, would it be worth discussing why? What processes would explain this? This implies more optimistic scenarios of ice cap preservation. Nevertheless, a model predicting less responsive ice cap mass balance decrease is actually counterintuitive. The schematic in Fig. 5 depicts that without vertical displacement, the ice mass is vulnerable to increasingly negative mass balance. So what processes could explain the lower sensitivity seen in the model? Some clarity here would make the paper stronger.

As discussed in our response to the previous paragraph, we were comparing the effects of glacier altitudinal adjustments and not linear vs. nonlinear mass balance models. Therefore, we think this sentence is well aligned with the main message of the paper.

The physical reasons behind this increased sensitivity of linear mass balance models are discussed in the first big paragraph of the Discussion section (L238-287). Mass balance models are calibrated with past climate and mass balance data, for which shortwave radiation plays a more important role (or fraction) than in the future climates that we will experience in the mid-late 21st century. This is due to the fact that a large share of future increase in snow and ice melt at the glacier surface will come from an increase in longwave radiation and turbulent fluxes consecutively to increasing temperatures. Consequently, the combination of physical processes that explained mass balance in the past will not be the same as in the future. This means that mass balance sensitivity to atmospheric variables changes with climate, but since linear models cannot take this into account they can only keep stationary sensitivities, which are likely to be too high for future climates. However, this is well captured by a nonlinear mass balance model (deep learning), since it can produce a different sensitivity depending on the input climate data. By comparing the responses captured in Fig. 3 with the physical basis known from the literature, we prove that our model has captured a realistic response to climate changes, which serves to build our hypothesis and argument about the overly pessimistic mass balance projections for ice caps and overly optimistic projections for low emission scenarios for mountain glaciers.

I like the new Fig. 5. However, two other issues: First, I find myself mixed up about positive and negative feedbacks. Wouldn't the retreat to higher altitudes of the mountain glacier case be depicting a negative feedback (i.e. counter acting the nature of the initial perturbation, while the thinning ice cap is a positive (i.e. self-reinforcing), feedback?

We thank the reviewer for raising this non-trivial point. For the refactoring of the manuscript we have been talking to a lot of different people in order to acquire a good idea of what is the general perspective on the "sense/direction" of feedbacks and biases. What we found out is that it is very heterogeneous and there is not really a consensus about this. The way we perceive it it may be clearer to keep the signs of the feedbacks exactly as they are for the mass balance. Meaning that a positive feedback (i.e. response) to the mass balance signal implies a change towards less negative or more positive mass balances. Alternatively, a negative feedback implies a change towards more negative or less positive mass balances.

Given this existing ambiguity, we have decided to reformulate the way we refer to these feedbacks. We now refer to the overall process as "topographical feedback", but when mentioning a given direction we refer to it as either a "positive impact on MB" or a "negative impact on MB". This way, we translate it directly into absolute terms for the MB, which makes it more straightforward for the reader to understand.

Second, the figure states "nonlinear mass balance sensitivity" following "+extreme negative mass balance." Yet shouldn't the figure more clearly show that the model predicts less mass loss than the linear model?

This comment directly relates to the two first remarks on this second round of reviews. Two different analyses are performed in the manuscript: the linear vs nonlinear mass balance models and the role of glacier geometry and topographical feedback. The goal of Fig. 5 is to explain the consequences of combining these two analyses. First it derives the implications of the two main types of topographical feedback: mountain glaciers vs ice caps. The comment of "+ extreme negative mass balance" falls into that category, and therefore it is a consequence of the lack or reduced amount of topographical retreat of ice caps. The second (lower) part of the figure deals with the consequences of linear vs nonlinear mass balance models, which starts under the text "Implications for temperature-index models". In this section the implications are in the direction mentioned in this comment, indicating that linear mass balance models over-estimate mass losses for all climate scenarios for ice caps.

In order to make it clearer for the readers, we have updated Fig. 5 with "impacts on mass balance" instead of "feedback" and a line has been added to separate the upper part to the consequences section.

Minor stylistic points: It might make the discussion easier to read by breaking up that first immense paragraph (i.e. end of L259).

That is a good point. We have split the paragraph in order to increase readability.

Is "parallelisms" a good word choice? I find it arcane, and reminds me of grade school grammar class.

Maybe it is not, not being native English speakers we might have borrowed that from our native tongues. We have now replaced it with “analogies”.

Overall, a very interesting and well written paper. Hopefully my ambiguities are merely my lack of understanding of the method.

Thank you again for your kind words. We are very pleased with this really constructive and detailed review.